# Controlled alignment of supermoiré lattice in double-aligned graphene heterostructures

Junxiong Hu [1,2,7], Junyou Tan[2,7], Mohammed M. Al Ezzi[1,2,7], Udvas Chattopadhyay[1,2], Jian Gou [1], Yuntian Zheng[1], Zihao Wang [3,4], Jiayu Chen[1], Reshmi Thottathil [1], Jiangbo Luo[1], Kenji Watanabe [5], Takashi Taniguchi [6], Andrew Thye Shen Wee [1], Shaffique Adam [1,2,3] & A. Ariando [1] ✉

The supermoiré lattice, built by stacking two moiré patterns, provides a platform for creating flat mini-bands and studying electron correlations. An ultimate challenge in assembling a graphene supermoiré lattice is in the deterministic control of its rotational alignment, which is made highly aleatory due to the random nature of the edge chirality and crystal symmetry. Employing the so-called "golden rule of three", here we present an experimental strategy to overcome this challenge and realize the controlled alignment of double-aligned hBN/graphene/hBN supermoiré lattice, where the twist angles between graphene and top/bottom hBN are both close to zero. Remarkably, we find that the crystallographic edge of neighboring graphite can be used to better guide the stacking alignment, as demonstrated by the controlled production of 20 moiré samples with an accuracy better than ~ 0.2°. Finally, we extend our technique to low-angle twisted bilayer graphene and ABC-stacked trilayer graphene, providing a strategy for flat-band engineering in these moiré materials.

The moiré superlattice, created by stacking van der Waals (vdW) heterostructures with a controlled twist angle[1–6], enables the engineering of electronic band structures and provides a platform for investigating exotic quantum states, both in the weakly interacting electron systems[7–10], as well as recently in the strongly correlated electron systems[11–17]. Particularly, when two moiré superlattices contact and interface together, the overlay of double moiré will create a new structure called supermoiré lattice, strongly modifying the lattice symmetry and electronic band structure[18–22]. Compared with the single moiré potential, the double moiré potential will further break the lattice symmetry and create isolated flat moiré minibands, providing a strategy for the band flattening effect[23]. Recently, evidence of possible

correlated states was observed in the double-aligned graphene supermoiré lattice[24], which triggered further effort to search for other correlated phenomena, such as superconductivity and ferromagnetic states, as observed in twisted graphene systems[25]. However, due to the sophisticated stacking and the lack of control of rotational alignment, searching for these correlated phenomena in double-aligned supermoiré lattice remains elusive.

In previous studies of double-aligned hexagonal boron nitride/graphene/hexagonal boron nitride (hBN/G/hBN) supermoiré lattice, several techniques have been developed to control the rotation alignment, such as in situ rotation mediated by atomic force microscope (AFM) tips and polydimethylsiloxane (PDMS) hemisphere[20,21].

[1]Department of Physics, National University of Singapore, Singapore 117542, Singapore. [2]Centre for Advanced 2D Materials and Graphene Research Centre, National University of Singapore, Singapore 117551, Singapore. [3]Department of Materials Science and Engineering, National University of Singapore, Singapore 117575, Singapore. [4]Institute for Functional Intelligent Materials, National University of Singapore, Singapore 117544, Singapore. [5]Research Center for Functional Materials, National Institute for Materials Science, Tsukuba 305-0044, Japan. [6]International Center for Materials Nanoarchitectonics, National Institute for Materials Science, Tsukuba 305-0044, Japan. [7]These authors contributed equally: Junxiong Hu, Junyou Tan, Mohammed M. Al Ezzi. ✉e-mail: ariando@nus.edu.sg

However, these techniques are restricted by either specialized equipment or complicated sample preparation steps. Consequently, the optical alignment of straight edges is still the most popular and reliable technique for moiré device fabrication[8–10,14–19,22–30]. Nevertheless, the conventional optical alignment has two limitations. First, the lack of prior understanding of the crystallographic orientation of graphene or hBN, leading to only a 50% and 25% success rate for alignment of a single and double moiré structure, respectively, because the zigzag (armchair) edge of graphene can be unintentionally aligned to either the zigzag or armchair edge of hBN. Moreover, the lattice symmetry of hBN layer also obscures the inversion symmetry of the moiré heterostructure[20,21], further decreasing the success rate to 12.5% in doublyaligned hBN/graphene/hBN heterostructures. Second, the optical alignment highly depends on the crystal edge of the graphene itself, and usually, it has a short and non-perfect straight edge. This can lead to an error in the representation of the actual principle crystallographic axes (PCA) during alignment.

In this work, we overcome the above two limitations and realize the control alignment of double-aligned graphene supermoiré lattice. First, we use a 30° rotation technique to control the alignment of top hBN and graphene, while we use a flip-over technique to control the alignment of top hBN and bottom hBN. Based on these two techniques, we can control the lattice symmetry and tune the graphene band structure. In a high-quality perfect double-aligned device with mobility of ~700,000 cm²/Vs at 2 K, we observe sharp resistive peaks at band fillings of 0, −4, −8 electrons per moiré unit cell, consistent with our calculated band structure. Second, we show that the neighboring graphite edge can be used to better guide the alignment, as demonstrated by the controlled production of 20 moiré samples with accuracy better than ~0.2°. Moreover, we have developed a so-called "Golden Rule of Three" that further guarantees the success rate and precision of our technique. Finally, we extend our alignment technique to other strongly correlated electron systems, such as low-angle twist

bilayer graphene and ABC-stacked trilayer graphene, enabling us to examine moiré potential effects in these strongly correlated electron systems.

## Results
### 30° rotation technique
First, we study the controlled alignment between top hBN (T-hBN) and graphene. In the optical alignment technique, the G/hBN moiré superlattice is achieved by aligning the PCA between graphene and hBN. Based on the crystallographic structures, they can form two basic moiré patterns: 0° G/hBN and 30° G/hBN (Fig. 1a, also Supplementary Fig. 1). When in the assembly of hBN/graphene/hBN sandwich structure, there are eight possible configurations: C1 (0°/0°) & C1* (0°/60°), C2 (0°/30°) & C2* (0°/90°), C3 (30°/30°) & C3* (30°/90°) and C4 (30°/0°) & C4* (30°/60°) (Fig. 1a). All possible configurations lead to 1/2 (50%) success rate for single alignment (C2, C2*, C4, C4*), and 1/8 (12.5%) success rate for double alignment (C1). Our theoretical calculations investigate the adhesive energies of two basic moiré patterns and find that there are two energy minima at 0° and 30° twist angles, indicating these two moiré patterns are energetically stable states (Fig. 1b, c). These calculations also suggest that graphene and hBN tend to self-rotate to 0° or 30° when stacked together, depending on the initial states[31,32].

Based on these calculations, we develop the so-called "30°-rotation technique" to simultaneously obtain these two energy-stable states. The central concept of this technique is illustrated in Fig. 1d–f. Instead of directly picking up the whole graphene as in the conventional optical alignment, the hBN is aligned partially with a graphene layer (Fig. 1d). Thanks to a stronger vdW interaction between graphene and hBN, the graphene layer can be torn into two pieces. The graphene area in contact with hBN can be selectively detached (G1), leaving behind a section of the graphene layer (G2) on the silicon wafer. The critical step in our technique is that the left graphene (G2) is

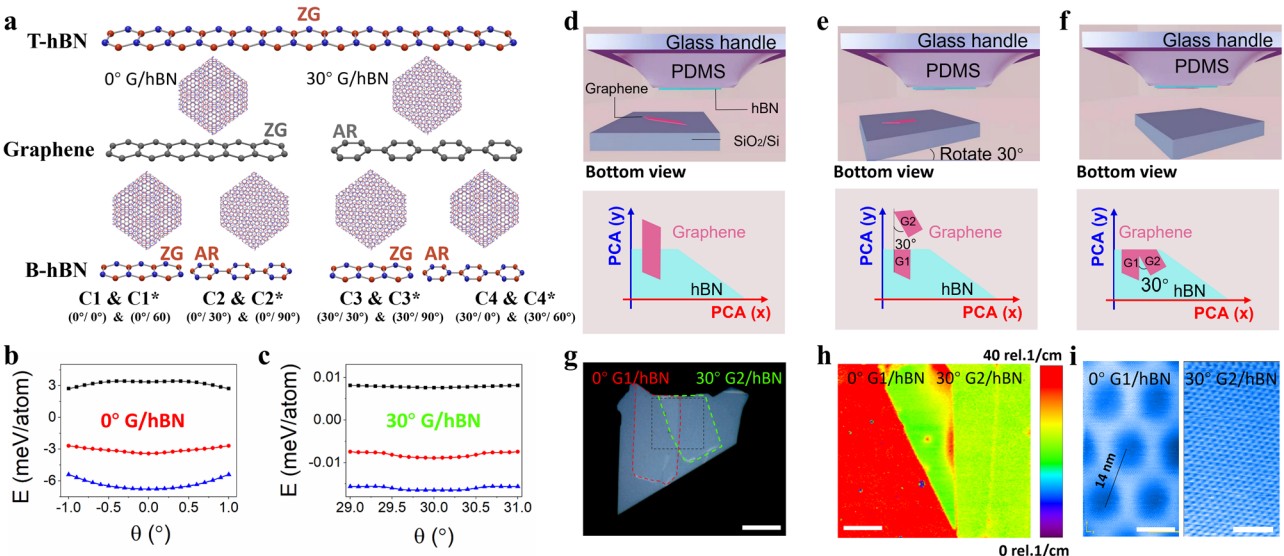

**Fig. 1 | Control alignment of top hBN and graphene by rotating 30°. a** Alignment of top hBN (T-hBN), graphene and bottom hBN (B-hBN). The Zigzag (ZG) edge of T-hBN is aligned with the ZG edge or Armchair (AR) edge of graphene and is then aligned with the ZG or AR edge of B-hBN, leading to eight possible combinations: C1 (0°/0°) & C1* (0°/60°), C2 (0°/30°) & C2* (0°/90°), C3 (30°/30°) & C3* (30°/90°) and C4 (30°/0°) & C4* (30°/60°), where C (or C*) represents the configuration when the T-hBN and B-hBN have the same (or opposite) lattice symmetry. The middle cartoons are two basic moiré patterns of 0° G/hBN and 30° G/hBN. **b, c** Calculated interaction energies for G/hBN heterostructure around 0° and 30°. Total energy (red circles) contributions from intralayer (elastic energy/blue triangles) and interlayer interactions (adhesive energy/black squares). **d–f** Side view and bottom

view of 30° rotational alignment. PCA refers to the principle crystallographic axes of crystals. G1 and G2 refer to graphene 1 and graphene 2 which come from the same flake. **g** Optical image of G/hBN stack on Polydimethylsiloxane (PDMS) stamp. The red dash line profiles the outline of 0° G1/hBN and the green dash line profiles the outline of 30° G2/hBN. Scale bar, 20 μm. **h** Spatial map of the full width at half maximum (FWHM) of 2D-band for the dashed area in (**g**). The red color map refers to 0° G1/hBN, while the green color map refers to 30° G2/hBN. Scale bar, 5 μm. **i** The STM topography image of 0° G1/hBN shows ~14 nm moiré patterns (left, 500 mV,15pA), and the topography image of 30° G2/hBN shows the characteristic of quasicrystal (right, 100 mV, 100pA). Scale bar, 10 nm.

rotated manually by a twist-angle of 30° (Fig. 1e), and G2 is then stacked at another location on the same hBN (Fig. 1f). This process results in two G/hBN structures based on the same hBN layer: G1/hBN and G2/hBN. Since G1 and G2 are 30° rotated to each other, one of them must be 0° aligned with hBN, and the other one must be 30° with hBN (Supplementary Fig. 1). To examine our concept, we investigate the resulting G/hBN structures using optical microscopy, Raman spectroscopy and scanning tunneling microscopy (STM). Figure 1g shows the optical image of G/hBN stacks, showing two sections of graphene rotated 30° to each other as indicated by the dashed lines. Figure 1h shows the respective full width at half maximum (FWHM) mapping of Raman 2D peaks. The FWHM mapping of the red area is close to 40 cm$^{-1}$, indicating the 0° rotation between graphene and hBN[33] (Supplementary Fig. 2). While the FWHM mapping of the green area is close to 20 cm$^{-1}$, indicating the 30° G/hBN. Moreover, the homogeneous distribution mapping indicates the spatially uniform twist angles. To further confirm the twist angles, we also use STM to directly characterize the moiré patterns of G1/hBN and G2/hBN (Fig. 1i). The 0° G1/hBN has a clear moiré wavelength of ~14 nm[34,35], while the 30° G2/hBN shows the character of quasicrystal line, which has 12-fold rotational order but lacks translational symmetry[36,37] (Supplementary Fig. 3). Moreover, the twist angles are also confirmed by our transport measurements, as discussed later. Therefore, using this 30°-rotation technique, we can always obtain the 0° G/hBN without the need to consider the exact chirality edge of each layer.

## Using neighboring graphite edge

Even though we can overcome the uncertainty in the edge chirality, there are still two other challenges for optical alignment. First, it is incredibly challenging to find a single-layer graphene flake with a straight edge, which usually happens in <1% from all exfoliated flakes (Supplementary Fig. 4). Second, the edge of single-layer graphene usually is not long and straight enough, decreasing the accuracy in the

alignment (Supplementary Figure 5). To improve productivity and accuracy, we show that the neighboring graphite edge can be better for alignment. The concept is illustrated in Supplementary Fig. 6. Figure 2 illustrates three typical cases of neighboring graphite edges that can be used for perfect alignment. The first case is that the single-layer graphene has a direct connection with its neighboring graphite edge (Fig. 2a). In this case, the single-layer graphene must share the same PCA with neighboring graphite edges, which means the PCA of graphite can be used for alignment. Our Raman 2D-band mapping confirms one part belongs to 0° G/hBN and the other to 30° G/hBN, as the FWHM of the 2D peaks is 40 and 20 cm$^{-1}$, respectively (Fig. 2c). The second case is that the single-layer graphene has no direct connection with the graphite edge, but one edge of graphene is multiples of 30° with the graphite edge. In this case, the graphene and the neighboring graphite also share the same PCA (Supplementary Fig. 7). Therefore, the PCA of graphite can be used for alignment. Raman 2D-band confirms the alignment since they have 0° G/hBN and the second part has 30 °G/hBN (Fig. 2f). The third case is that the single-layer graphene neither has any connection nor is multiples of 30° with the neighboring graphite edge (Fig. 2g), we show that the PCA of neighboring graphite can still be used for alignment, since our Raman spectra show that one part is 0° G/hBN and the second part is 30° G/hBN (Fig. 2i). Regarding the distance between graphene and graphite, we have studied three different cases with a distance of 40, 130 as well as 250 μm (Supplementary Fig. 8). We found that in all three cases, the neighboring graphite edge can be used for alignment if we use the so-called "non-overlap exfoliation" method (Supplementary Fig. 9).

The best advantage of our technique is that the alignment is not limited by the geometry of graphene itself, and any random shape of single-layer graphene flake can be used for fabricating moiré samples, as long as its neighboring graphite can provide the straight edges which are nominally sharing the same termination as reference points (Supplementary Fig. 10). In order to demonstrate the high productivity

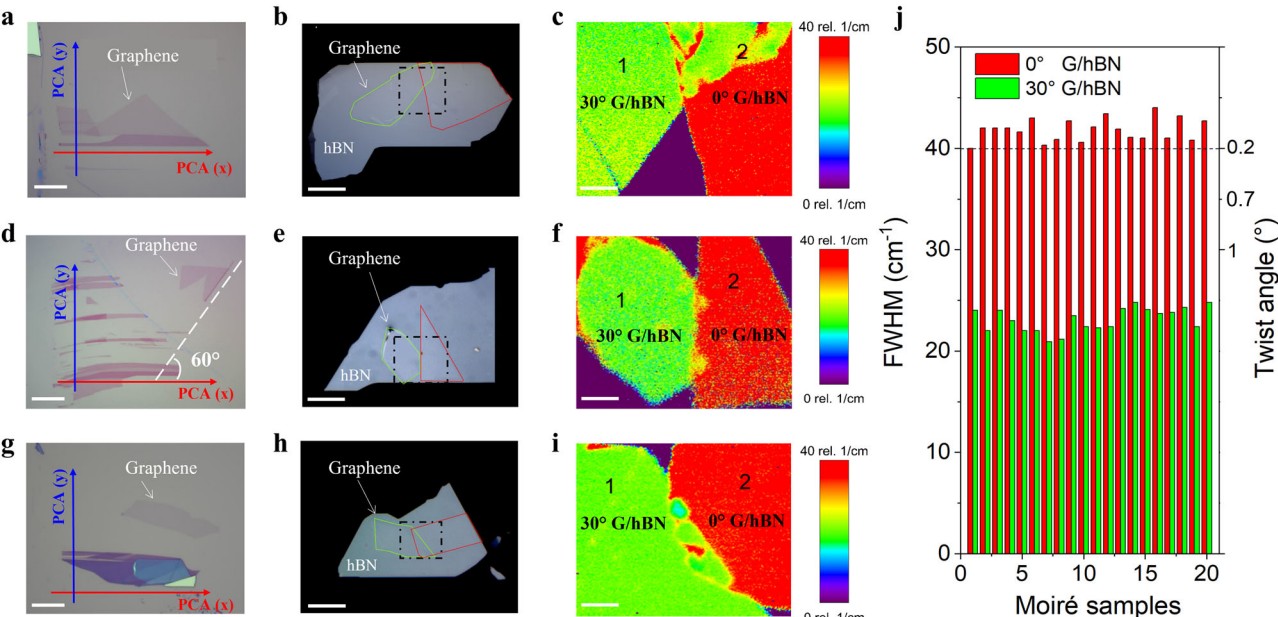

**Fig. 2 | Perfect alignment of top hBN and graphene using the neighboring graphite edge. a** Optical image of single-layer graphene connecting with a neighboring graphite edge. **b** G/hBN stack after alignment using the graphite edge of (**a**). The red line profiles the outline of 0° G/hBN and the green line profiles the outline of 30° G/hBN. **c** Spatial map of the FWHM of Raman 2D-band for the black dashed area in (**b**). **d** Single-layer graphene with one edge (white dashed line) has 60° with a neighboring graphite edge. **e** G/hBN stack after alignment using the graphite edge of (**d**). **f** Spatial map of the FWHM of Raman 2D-band for the black dashed area in

(**e**). **g** Single-layer graphene without any connection with an adjacent graphite edge. **h** G/hBN stack after alignment using the graphite edge of (**g**). **i** Spatial map of the FWHM of Raman 2D-band for the black dashed area in (**h**). Scale bars, 20 μm (**a**, **d**, **g**); 5 μm (**b**, **e**, **h**); 2 μm (**c**, **f**, **i**). **j** Histogram of the FWHM of Raman 2D-band and twist angle for 20 moiré samples. The FWHM of ~20 cm$^{-1}$ (green color) corresponds to 30° G/hBN, and the FWHM of ~40 cm$^{-1}$ (red color) corresponds to 0° G/hBN. The FWHM of 20 moiré samples is larger than 40 cm$^{-1}$, as indicated by the horizontal dashed line, indicating that the accuracy of our technique is better than ~0.2°.

and accuracy of our strategy, we further fabricated 20 moiré samples, where all the single-layer graphene flakes have a random edge. The histogram shows that the FWHM of all the samples is larger than 40 cm$^{-1}$, indicating that the accuracy of our alignment is better than ~0.2°. (Fig. 2j, see also Supplementary Fig. 11). In comparison, the alignment accuracy when using a conventional technique is merely 0.5- 1° (See the summary in Supplementary Table 1). Compared with single-layer graphene alignment, we find that using the neighboring graphite edges can be better for alignment because the thick graphite edge can have a longer and more straight edge. Moreover, following the "Golden Rule of Three" (see discussion section for details) when using the neighboring graphite edges further guarantees the success rate and precision of our technique. Our results show that a straight graphite edge can always ensure the alignment better than 0.2°, while a non-perfect straight edge or short straight edge (5–10 $\mu m$) will lead to a significant deviation from an ideal alignment (>0.5°) (Supplementary Fig. 5). Therefore, using the neighboring graphite edge for alignment, we can not only significantly improve the device yields, but also guarantee the high accuracy of alignment.

### Flip-over technique

Next, we study the control alignment of T-hBN and B-hBN. Because of the uncertain edge chirality of B-hBN, there are again two possible cases when we stack the 0° and 30° T-hBN/G on the B-hBN: C1/C3, if T-hBN and B-hBN have the same edges (Supplementary Fig. 12) and C2/C4, if T-hBN and B-hBN have the different edges (Supplementary Fig. 13). To secure the crystallographic orientation of T-hBN and B-hBN, we can use the same edge of the same hBN for alignment. However, apart from the edge chirality, we also need to consider the lattice symmetry of each hBN layer. As shown in Fig. 3a, for the conventional pick-up process, we pick up BN1 and stack it on BN2, so the bottom surface of BN1 is in contact with the top surface of BN2. Depending on the surface symmetry determined by the layer number of hBN, the final stack can be C1 (0°/0°) or C1* (0°/60°) (Fig. 3b, c). The C1 (0°/0°) heterostructure has three-fold rotationally symmetric, and the overall structure breaks inversion symmetry, while the C1* (0°/60°) heterostructure has a six-fold rotationally symmetric and the structure host

inversion symmetry. Even though these two structures have the same moiré wavelength, the change of local stack induces a change in the atomic relaxation, consequently leading to totally different bandgaps[20,23].

In order to control the alignment of T-hBN and B-hBN, we then develop a "flip-over technique" based on the same crystal edge as well as the same atomic surface of hBN. As illustrated in Fig. 3a, the key step is that BN2 is flipped over before placing BN1 on BN2. In this case, the same lattice symmetry of T-hBN and B-hBN can be guaranteed. We prefer to choose the bottom surface of the hBN instead of the top surface because the two disjoint sections of the hBN may have a different number of layers, and their top surfaces might not be atomically flat and clean. On the other hand, however, it is highly likely the bottom surfaces of the hBN flakes have the same termination and are much cleaner, as the bottom surfaces of the two hBN flakes are cleaved from the same crystallographic facets of hBN crystal and are not in direct contact with the tape used for the cleaving. Further, the difference in thickness should not affect the termination of the bottom surface. Therefore, the main aim during the flip-over step is to attach the bottom surface of one of the flakes onto the bottom surface of the other flake (both bottom surfaces should have the same termination) instead of combining both top surfaces.

Based on this concept, we can consider three different routes for obtaining BN1 and BN2 to be used in the flip-over technique (Supplementary Fig. 14). First, one hBN can be cut into two (BN1 and BN2). In this case, we can make sure that BN1 and BN2 share not only the same PCA, but also the same surface. However, this cutting process is tedious, requiring complicated lithography steps. Second, BN1 and BN2 can come from naturally fractured hBN flakes, which can always be found during mechanical exfoliation. Figure 3d shows the optical images of two pieces of fractured hBN that can be regarded as T-hBN and B-hBN, respectively. Combining the 30° rotation and flip-over technique, we can first realize the single alignment and then the double alignment, as demonstrated by the increase in FWHM of Raman 2D-band from ~40 cm$^{-1}$ (single alignment) to ~70 cm$^{-1}$ (double alignment) at the same area[19] (Fig. 3e, see also Supplementary Fig. 15). Third, we can also obtain BN1 and BN2 from two neighboring hBN flakes whose

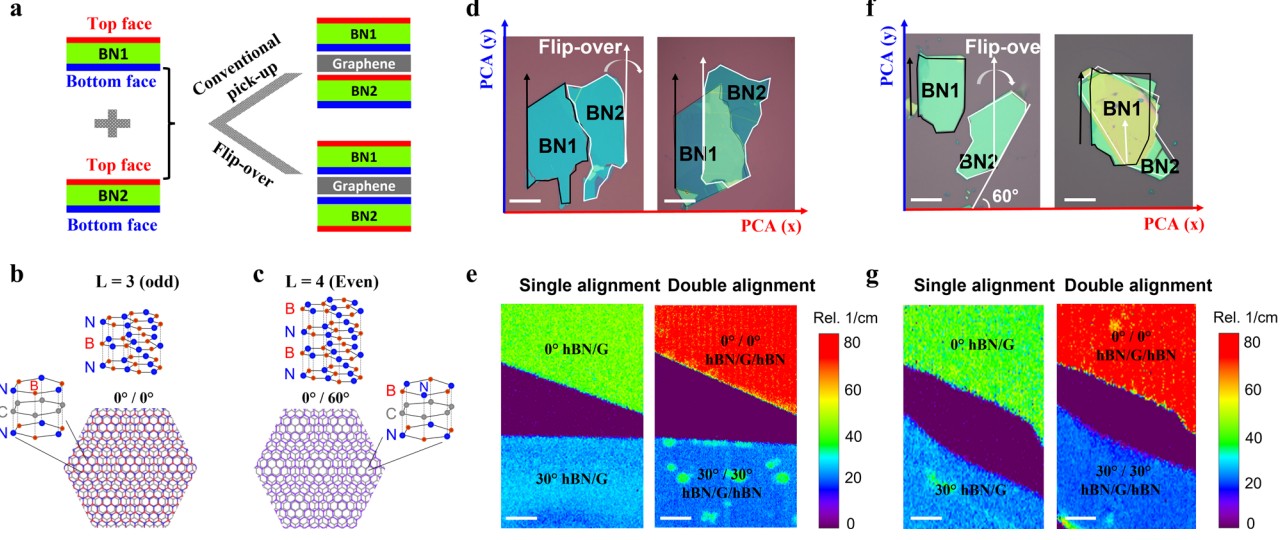

**Fig. 3 | Control alignment of top hBN and bottom hBN using the neighboring hBN surface. a** Schematics of conventional pick-up and flip-over technique for double alignment. **b, c** Schematics of hBN/graphene/hBN heterostructures with odd (**b**) and even (**c**) hBN layers. Lattice models at the high symmetry points of the moiré pattern show the atomic arrangement for each. Purple shading in (**c**) denotes the overlap of boron (red) and nitrogen (blue) in the T-hBN and B-hBN. **d** Optical images of fractured hBN and alignment of T-hBN and B-hBN using the flip-over

technique. The black lines profile the T-hBN, and the red lines profile the B-hBN. **e** Maps of the FWHM of Raman 2D-band of graphene for the single and double alignment using the hBN in (**d**). **f** Optical images of two neighboring hBN and one of hBN have 60° with PCA can also be aligned using the flip-over technique. **g** Maps of the FWHM of Raman 2D-band of graphene for the single and double alignment using the hBN in (**f**). Scale bars, 10 $\mu m$ (**d**, **f**); 1 $\mu m$ (**e**, **g**).

straight edges have integer multiples of 30 degrees to each other, as shown in Fig. 3f, then the two adjacent hBN flakes have a high chance of coming from the same crystal (Supplementary Fig. 14). In this case, we can also realize a perfect double alignment based on the same surface of hBN, as confirmed by the Raman 2D-band shown in the same area (Fig. 3g, see also Supplementary Fig. 15).

Therefore, combining the 30° rotation and flip-over technique, we can overcome the uncertainty in the edge chirality and lattice symmetry during rotation alignment, and realize the 100% success rate for the alignment of the double moiré structure. This is in stark contrast to the conventional technique that can only lead to a 12.5% success rate[24].

### Electronic transport measurements

Finally, to reveal the role of moiré potential in the reconstruction of lattice symmetry and band structure, we study the electronic properties of C1–C3 configurations by top-gate devices[38,39] (Fig. 4a, b). We first use Raman to verify the sample twist angle, and typical Raman data of a perfectly double-aligned device is shown in Fig. 3e, g. Subsequently, a standard lithographic technique is used to pattern a Hall bar geometry. For double-aligned C1 (0°/0°) device, apart from the charge neutrality point (CNP), there are also two successive satellite peaks that appear at hole-side, locating at $-n_s$ and $-2n_s$, where $n_s = 2.3 \times 10^{12}$ cm$^{-2}$ (Fig. 4c). Since $n_s$ is the carrier density required to reach the edge of its first Brillouin zone of a moiré pattern $\lambda$ by $n_s = \frac{8}{\sqrt{3}\lambda^2}$[8-10], we can calculate the moiré wavelength of C1 of ~14 nm, in good agreement with our Raman and STM measurements. Moreover, our band structure calculation (Inset of Fig. 4c) shows that the double moiré superlattice significantly splits the hole-sided bands at higher energy into minibands with fully developed gaps at electron fillings of $-n_s$, $-2n_s$ and $-3n_s$, which is consistent with our observation since these satellite peaks in hole-side bands are more developed than in the electron-side. Moreover, apart from these band filling peaks, we do not observe any extra peaks which

may come from the misalignment[18-24]. Therefore, we can conclude that the C1 sample is a perfect double-aligned sample, where the twist angles between graphene and top/bottom hBN are both close to zero. A similar feature is also observed in the second perfect double-aligned sample (Supplementary Fig. 16). While for single-aligned C2 (0°/30°) device, we only observe one satellite peak at hole-side, locating at $-n_s = -2.2 \times 10^{12}$ cm$^{-2}$, corresponding to the moiré wavelength of ~14.5 nm (Fig. 4d). The slightly larger moiré periodicity in C2 can be attributed to the strain effect in the heterostructure[19,24]. The lack of $-2n_s$ peak in C2 suggests that the overlap between the second and third band, as our band structure calculation shows a rather small bandgap (Inset of Fig. 4d). When the moiré potential disappears in C3 (30°/30°) device, there is only one main Dirac point, consistent with our band structure calculation (Inset of Fig. 4e).

When the electrons simultaneously subjected to both a magnetic field and a spatially periodic electrostatic fields, the energy spectrum develops into a Landau fan with a fractal structure known as the Hofstadter butterfly[8-10], which can be renormalized into a diagram defined by the Diophantine equation: $\frac{n}{n_0} = \nu \frac{\phi}{\phi_0} + s$, where $\nu$ is the Hall conductivity in units of a conductance quantum $e^2/h$, as indicated by black solid lines in Fig. 4f, g, h, with topological index $\nu$ of $\pm 2, \pm 6, \pm 10, \pm 14,...,$ and $s$ the index of band filling. $\phi = B \cdot A$ is flux per moiré unit area at magnetic field B, and $\phi_0 = h/e$ is a flux quantum with $h$ being the Planck's constant. Figure 4f shows the fracture spectrum of C1 (0°/0°) device. The straight minigaps arises at $\frac{\phi}{\phi_0 q}$ ($q = 1, 2, 3, ...$), resulting in Brown-Zak (BZ) oscillations with the fundamental period field of $B_f = 24.49$ T ($\phi_0/A$). Thus, we can calculate the moiré periodicity of D1 of 13.97 nm, which is consistent with the first band filling density at $-n_s = -2.35 \times 10^{12}$ cm$^{-2}$. Similarly, the BZ oscillations for C2 device has the $B_f = 23$ T (Fig. 4g). We can then calculate the moiré periodicity of C2 to be 14.417 nm, also consistent with the first band filling $-n_s = -2.2 \times 10^{12}$ cm$^{-2}$. Apart from the integer filling factions of

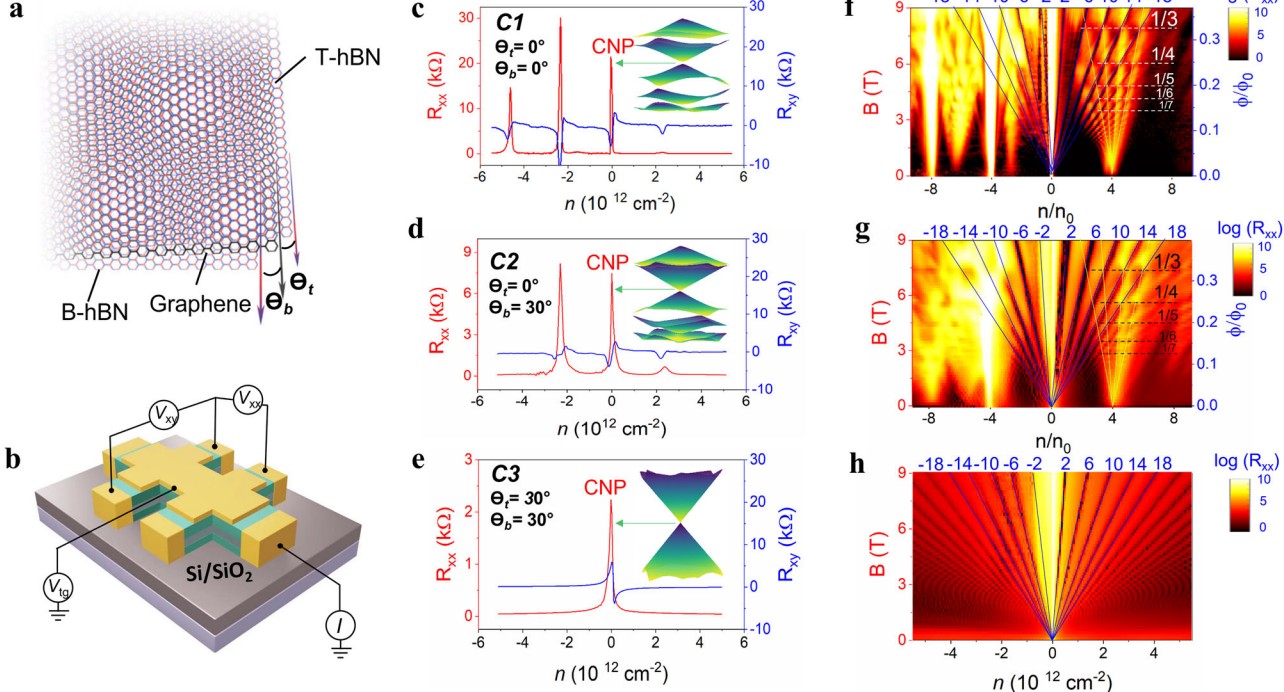

**Fig. 4 | Lattice symmetry and band structure tuned by moiré potential in top hBN/graphene/bottom hBN heterostructure. a** Art view of the supermoiré lattice with twist angles ($\theta_t$ and $\theta_b$) between graphene and T-hBN and B-hBN. **b** Schematics of the top-gate device with double moiré. Longitudinal resistance (Left axis) and Hall resistance (right axis) with $B = 0.5$ T versus carrier density for (**c**), C1 (0°/0°), (**d**), C2 (0°/30°), and (**e**), C3 (30°/30°). The insets show the corresponding band structures at the K-point. CNP refers to the charge neutrality point of Dirac band. Landau fan diagram of (**f**), C1, (**g**), C2 and (**h**), C3 plotted in magnetic field (left) and corresponding $\phi/\phi_0$ versus $n/n_0$. $\phi/\phi_0$ and $n/n_0$ are the normalized magnetic flux and carrier density, respectively. The top number are the topological index $\nu$ to be $\pm 2, \pm 6, \pm 10$, etc. $T = 2$ K.

0, − 4, − 8 in the Landau fan diagram, there are also some anomalous features between them, which can be ascribed to the low energy Van Hove singularities. Similar phenomena were previously observed in single-aligned devices by transport study[40] as well as the optical study[41]. Finally, when the moiré periods disappear in C3 device, the BZ oscillations also disappear, as there is only the Landau levels spectrum (Fig. 4h). These results suggest that the control alignment of moiré patterns, acting as a periodic moiré potential, can tune the lattice symmetry and engineer the band structure.

## Discussion

In this work, we overcome the uncertainty of the edge chirality and crystal symmetry by using 30° rotation technique and flip-over technique, and finally realize the control alignment of double-aligned graphene supermoiré lattice. Moreover, we show that neighboring graphite edges can be used to better guide the stacking alignment. A successful alignment of graphene and hBN relies on two things: (1) Precise representation of the PCA before alignment, and (2) Correct alignment techniques as reported in our main text. Since graphene and hBN come from two different flakes, there is a challenge to precisely represent the PCA of each flake. We have overcome this challenge and set several rules, the so-called "Golden Rule of Three", that have to be strictly followed to guarantee the alignment precision and success rate. The rules are as follows:

*Golden Rule 1*: Do not use the edge of graphene itself. Instead, use the straight edge of the neighboring graphite. Moreover, the length of the straight edge should be no less than 100 µm (Supplementary Fig. 17a–c).

*Golden Rule 2*: Do not use a graphite flake with a single straight edge. Instead, use a graphite flake with multiple straight edges that are offset by integer multiples of 30 degrees to each other (Supplementary Fig. 17d–f).

*Golden Rule 3*: Do not accept random fluctuation of angle measurements larger than 0.2 degree. Instead, aim to minimize fluctuation by measuring angle multiple times when representing the PCA on the optical microscopy system (Supplementary Fig. 17g–i).

Below we explain why we must strictly follow the "Golden Rule of Three".

First, we do not suggest using the edge of graphene, because the graphene has a poor contrast under optical microscopy, making it difficult to represent the PCA. On the contrary, the edge of thick graphite is more obvious. Moreover, in order to keep the high precision of 0.2 degrees for alignment, the error of the PCA should be controlled within 0.2 degrees. Therefore, we must choose a graphite flake with a long straight edge. The longer the edge, the smaller the error is. Supplementary Fig. 18 demonstrates how the length of the graphite edge affects the angle measurement. When the length of the edge is smaller than 50 $\mu m$ (Supplementary Fig. 18a–c), the random error of PCA is larger than 0.5 degrees even with three measurement repeats. In this case, it is impossible to realize precise alignment below 0.2 degrees. Thus, we can also explain why, in earlier reports, the misalignment is usually as large as 0.5–1 degree, as summarized in Supplementary Table 1. On the other hand, if the length of the edge is more than 100 $\mu m$ (Supplementary Fig. 18d–f), the random error can be controlled well below 0.2 degrees. In this case, it is possible to achieve the precise alignment of below 0.2 degrees, as shown in Fig. 2j. The Golden Rule #1 thus dictates that we must use graphite flakes with a straight edge of more than 100 $\mu m$. Second, it is very common to get disordered edges that are an admixture of zigzag and armchair terminations. Therefore, we cannot represent PCA with 100% certainty if only depending on one single edge. Our Golden Rule #2 thus dictates that we must use graphite flakes with multiple straight edges aligned at integer multiples of 30 degrees. Third, when utilizing an optical microscope (in our case Nikon-LV100NDA) to identify and measure the angle of the graphite edge, the measured angles are not exact and prone to error (that can

be easily larger than 0.2 degrees) due to human error and the limited optical microscope resolution. When the fluctuation of the measured angles is more than 0.2 degrees, keeping the alignment precision between graphene and hBN as good as 0.2 degrees will be impossible. Our Golden Rule #3 thus dictates that we must measure the angle of the edges at least three times and ensure the angle fluctuation is below 0.2 degrees. Strictly following the above three golden rules and three main alignment techniques, we can remove the 1/8 (12.5%) limitation, allowing a high yield of close to 100% to be realized for the double-aligned hBN/graphene/hBN heterostructure. Moreover, the twist angle between each layer can be controlled well below 0.2 degrees. Furthermore, our technique can greatly improve the efficiency of fabricating samples. There is a clear improvement in the sample yield, precision and fabrication time using our technique, as we summarized in Supplementary Table 3.

In conclusion, we develop a generic strategy to overcome the edge chirality and lattice symmetry uncertainty in rotation alignment. Moreover, we show that the neighboring graphite edge can be used for better alignment of the stacking structures, significantly improving the device yield and alignment accuracy. Compared with previous conventional techniques, the current technique is easier and reliable to operate with robust and definite control of alignment. Considering the emerging area of "twistronics", our technique can be beneficial for much effort in this area in many laboratories. For example, our strategy can also be applied to the family of transition metal dichalcogenide (TMD) semiconductors[3,4] such as $WSe_2/WS_2$ moiré superlattice, where a prior measurement of edge chirality in each crystal is necessary before alignment[42–45]. To show the universality of our technique, we also extend our technique to other correlated systems, like low-angle twisted bilayer graphene (Supplementary Fig. 19) and ABC-stacked trilayer graphene (Supplementary Fig. 20). We believe our technique can help effort in exploring the physics of strong electronic correlations and non-trivial band topology in these moiré materials.

## Methods

### Fabrication of devices and characterizations

For sample preparation, the general process is as follows. We use polycarbonate (PC), or poly-propylene carbonate (PPC) film mounted on a thick PDMS stamp to move and orientate the flakes. The stamp is used to pick up the first top hBN layer. The hBN is then positioned and aligned to a graphene edge or neighboring graphite edge before the two are brought into contact. We quickly lift the stamp once the hBN is fully passed across the graphene. After this, we invert the stamp and perform various characterizations. We then pick up a bottom hBN layer and repeat our characterizations. Last, the stack of crystals is positioned and brought into contact with a silicon wafer at the PC melting point of 180 °C. The membrane is then removed slowly so that all the stacks are left on the silicon wafer. The standard Hall bar geometry of the devices is shaped by electron-beam lithography and etched by $CHF_3/O_2$ plasma. The edge contact electrodes and top electrodes (3 nm Cr/65 nm Au) are deposited by standard electron-beam evaporation. The carrier mobility of our device is calculated from the slopes of conductivity $\sigma(V_g)$ at a small concentration of $n$ $\sim 10^{11}$ cm$^{-2}$.

### Details of flip-over technique

The flip-over technique includes three steps. First, we used a thin film of polycarbonate (PC, Sigma-Aldrich, 6% dissolved in chloroform purchased at HQ Graphene) and polydimethyl-siloxane (PDMS) stack on a glass slide to pick up the first piece of hexagonal boron nitride flake (BN1) at 60 °C. Then we used the van der Waals force between BN1 and monolayer graphene to tear and pick up half of the graphene flake. The remaining graphene flake on the silicon was rotated by 30° and picked up at 40 °C. Second, the polypropylene carbonate (PPC) layer

was spun on top of a bare silicon wafer, released with a hollow-shaped Scotch tape, and then transferred on top of the second PDMS stack on a glass slide. The PPC (Sigma-Aldrich, CAS 25511-85-7) is made of propylene carbonate and anisole with a mass fraction of 5%. To enhance the interaction between PPC and PDMS, the PDMS is treated with oxygen plasma for 10 mins before being covered with PPC film. Then the PPC/PDMS stamp is used to pick up the second piece of hexagonal boron nitride flake (BN2) at 60 °C. Third, in order to expose the bottom surface of BN2, we flip over the PDMS/PPC/BN2 and make the bottom surface of BN2 to be exposed to the bottom surface of BN1. Finally, we use PC/BN1/G to pick up BN2 from PPC at 80–100 °C. The success of this procedure relies on the stronger adhesion between PC and BN1 compared to BN2 and PPC.

## The twist angle identified by transport

For transport measurement, the twist angles are estimated from two independent methods. First, we measure the gate voltages of full-filling gaps and convert these voltages to full-filling density $n_s$ using both the gate capacitance and Hall measurements. We then calculate the twist angle from which the full filling corresponds to four electrons per moiré unit cell so the moiré unit cell area $A = 4/n_s$. Second, we study Hofstadter's butterfly features under magnetic fields. Here we use carrier-density-independent oscillations, also called BZ oscillations, of the longitudinal resistance $R_{xx}$ fields, with the minimum of $R_{xx}$ under magnetic moiré unit cell of a fraction of the flux quantum, $BA = \varphi_0/N$, where B is magnetic field, $\varphi_0$ is the magnetic flux. The Hofstadter spectrum will exhibit fractal signature (i.e, B-8 T, B-6 T, and B-4.85 T) corresponding to $\varphi = \varphi_0/3$, $\varphi = \varphi_0/4$, $\varphi = \varphi_0/5$.

## The twist angle identified by Raman spectroscopy

Raman spectroscopy offers a simple and fast way to determine the twist angle. The laser wavelength is 633 nm with a power of 1 mW through a ×100 objective. High-resolution Raman maps are used to characterize the spatially unfirm twist angles, which are acquired with a scan parameter of eight points per micrometer over a 10–20 μm area. We use the full width at half maximum (FWHM) of the Raman 2D-band to estimate the twist angle, where the FWHM is analyzed by Gaussian fitting. According to the previous study [33], there is a linear dependence between $FWHM_{2D}$ and the moiré wavelength for twist angles below 2°, $FWHM(2D) \cong 5 + 2.6\lambda_M$ and $\lambda_M = \frac{1.018a_{CC}}{\sqrt{2.036[1-\cos(\theta)] + 0.018^2}}$. Therefore, the FWHM of the Raman 2D peak is 41.5 $cm^{-1}$ when the moiré wavelength is 14 nm at a perfect alignment of 0°.

## The twist angle identified by scanning tunnelling microscopy

For STM measurement, the graphene surface should be on the top so that it can be reached by STM tip. Therefore, we use a modified pick-up method for STM. First, we use a PC and PDMS stack on a glass slide to pick up a 20–30 nm-thick hBN flake. We then use the van der Waals force between hBN and graphene to pick up graphene. In order to expose the graphene surface at the top, the resulting stack with PC is transferred and released onto a second PDMS stamp. After dissolving the PC film in DCM solution, the inverted stack with PDMS is released on a $SiO_2$ wafer at 80 °C. After this, the Au/Cr electrode is evaporated for the electrical contact using a standard lithographic technique. Before inserting into the STM chamber, the G/hBN device is annealed at 300 °C for 3–5 h in ultrahigh vacuum to remove the surface contaminations.

Experiments are conducted with an Omicron LT-STM at low temperature (T = 77 K) with a base pressure better than $1 \times 10^{-11}$ mbar. Before the measurement, we check the tip by performing differential conductance (d$I$/d$V$) measurements on a clean Au (111) surface both before and after graphene measurement. d$I$/d$V$ spectra are measured using a lock-in technique with a 20 mV (r.m.s.) and 963 Hz modulation applied to the sample voltage.

## Data availability

Relevant data supporting the key findings of this study are available within the paper and the Supplementary Information file. All raw data generated during the current study are available from the corresponding authors upon request.

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

## Acknowledgements

This work is supported by the Ministry of Education (MOE) Singapore under the Academic Research Fund Tier 2 (Grant No. MOE-T2EP50120-0015), by the Agency for Science, Technology and Research (A*STAR) under its Advanced Manufacturing and Engineering (AME) Individual Research Grant (IRG) (Grant No. A2083c0054), and by the National Research Foundation (NRF) of Singapore under its NRF-ISF joint program (Grant No. NRF2020-NRF-ISF004-3518). S.A. and M.M.A.E acknowledge the support of the Singapore National Science Foundation Investigator Award (Grant No. NRF-NRFI06-2020-0003). K.W. and T.T. acknowledge support from the JSPS KAKENHI (Grant Numbers 19H05790, 20H00354 and 21H05233).

## Author contributions

A.A. conceived and supervised the project. J.X.H. designed and per-formed the experiments. J.Y.T. developed 30°-rotation alignment tech-niques and provided support and training on device fabrication process. M.M.A.E and U.C. carried out theoretical calculations under the super-vision of S.A. Y.T.Z., J.Y.C., R.T. and J.B.L. helped to prepare sample and make the device. Z.H.W. helped the data analysis and interpretation. T.T. and K.W. provided the bulk hBN crystals. J.G. helped STM measurement under the supervision of A.T.S.W. J.X.H. and A.A. analyzed the experi-mental data and wrote the paper. All authors discussed the results and commented on the paper.

## Competing interests

The authors declare no competing interests.
