## [Peer Review File · Nature Communications]

Controlled alignment of supermoiré lattice in double-aligned graphene heterostructuresEditorial Note: Parts of this Peer Review File have been redacted as indicated to remove third-party material where no permission to publish could be obtained.

REVIEWER COMMENTS

Reviewer #1 (Remarks to the Author):

By controlling the rotational alignment between adjacent Van der Waals crystals, one can build a heterostructure where the two lattice drift in and out of registry, forming a moire pattern. This additional periodicity profoundly modifies the electronic band structure. Such an effect was first realized in graphene aligned on hexagonal boron nitride (hBN), with the formation of secondary Dirac points at full-filling of the mini-Brilloin zone of the moire superlattice. However, repeatably forming such a moire superlattice of two different materials is experimentally challenging: one must precisely align two different crystals without any knowledge of the crystallographic axis. To date, this problem has been solved by rotationally aligning by eye long straight edges of exfoliated graphene/hBN which one assumes to be (approximately) well defined zig-zag or armchair terminations of the crystal lattices. However, without knowledge of the explicit termination, only half of fabricated samples will form a moire (assuming perfect alignment and perfect edges). An additional factor of two is lost if one is trying to align a graphene to both proximate hBN crystals in an encapsulated geometry.

The authors present a clever way to mitigate these ambiguities in how the crystals should be arranged. By applying the "rip and stack" technique used to fabricate twisted bilayer graphene, two sections of the same crystal are stacked at either zero or thirty degrees relative to a straight edge of the top hBN crystal. To aid in this alignment, adjacent graphite can be used as it is likely from the same progenitor crystal. In conjunction, the authors flip adjacent hBN crystals so that they can form a doubly aligned heterostructure where both the top and bottom graphene are aligned to the proximate hBN crystals. To conclude, they present transport measurements of such a device.

While I am excited by the results of this manuscript, there are substantial flaws that must be fixed before I can condone its publication. I detail my concerns below.

1. Discussions of technique, sample yield, and limitations

When discussing a new technique, it is extremely important to discuss the limitations. I found this discussion in the beginning of the manuscript very lacking. For example, when discussing the success rates for alignment of graphene/hBN heterostructures, the authors do not detail the key caveat of requiring the edges be perfectly crystalline. In graphene, it is very common to get disordered edges that are an admixture of zig-zag and armchair terminations. These edges can appear straight. One key way to rule this out is to look for multiple straight edges that are offset by integer multiples of 30 degrees.

Furthermore, the authors present a figure of merit of their technique of alignment of " ~ 0.2 degrees." The authors should be able to do a more rigorous assessment of their technique. Particularly, some figure of merit of alignment based on the lengths of the edges they aligned for the device. Additionally, supplementary tables S1 and S2 present the achieved angular alignment as 0 with no error bars.

When discussing possible configurations, there are eight possible configurations (assuming perfect alignment), as the top hBN can be either armchair or zig-zag in its termination (there are actually even more when one considers the c-axis termination of the hBN). This is incorrect in both the main text and figure 1.

Using adjacent graphite to align graphene is a clever technique and I applaud the authors in making many well aligned devices using this technique. However, the authors do not discuss any limitations, which are particularly interesting. Figure 2 shows that they are already succeeding in using graphite ~ 40 microns away to align graphene. Did any samples fail using this technique not come out as intended? What is the furthest away piece of graphite they have used for successful alignment? It certainly will not always be the case that adjacent graphene and graphite will share the same crystallographic axis.

hBN flipping

The discussion motivating the flipping of hBN is lacking. The authors do not mention that the hBN grown by NIMS is AA' stacking. Therefore, motivating the idea of trying to control the proximate hBN surface with the graphene by such a flipping technique. However, this appears to not be controlled at all as the two sections of hBN used in both the presented devices appear to be different colors (as seen by a crude check of their RGB values. even if they seemed to be the same color, I would not trust them to have the exact same number of layers as they are disconnected). There is then no control of inversion symmetry within the stacks.

2. Transport measurements

In discussing the Landau fans, it would be nice to have some comparison to other doubly aligned devices, as the authors claim they have achieved some of the best alignment. For example, Fig. 4F and Fig. 4G show fans emerging from integer number of chair carriers not equal to 4 or 8. These features can be ascribed to a simple Hofstadter picture but I am not sure if some of them have been so cleanly observed before.

3. Miscellaneous points

When plotting the Raman or transport data, do not use jet. It is misleading in that it provides substantial contrast to the eye where there should not be. It is technically most closely a divergent colormap and therefore not an appropriate choice for a strictly positive number anyway. Please use a perceptually uniform sequential colormap (<https://matplotlib.org/stable/tutorials/colors/colormaps.html>, this link also has interesting discussion on why such choices are important).

In the supplemental materials, while it is discussed elsewhere, the methods section does not detail the hBN flipping technique.

Reviewer #2 (Remarks to the Author):

In this article, the author presented a strategy to construct a double-aligned hBN/graphene/hBN moiré superlattice where crystallographic orientation of both top and bottom h-BN is aligned to that of graphene located in between h-BN.

A fabrication of double-aligned hBN/graphene/hBN is recognized as a challenging subject in the field of 2D materials, primarily due to the uncertainty of the edge chirality of graphene and h-BN. In my understanding, the author provided three experimental techniques to improve the reliability for fabricating double-aligned hBN/graphene/hBN. These are,

- 1) Use 30 degree rotated graphene flakes.
- 2) Using neighboring graphite edges to determine principal crystallographic axes of graphene.
- 3) Flip-over technique to control the alignment of top hBN and bottom hBN.

By using these techniques, the author achieved the fabrication of the following device.

- 4) Perfect double-aligned hBN/graphene/hBN moire superlattice.

These 1) to 4) can be regarded as a novelty of this manuscript.

The method 1), 2), and 3) is a good idea and I think these ideas were not reported before.

By using method 1), 2), and 3), the author improved the fabrication yield of double-aligned hBN/graphene/hBN devices. This may be a good improvement. Since this method still relied on the optical microscopy image of the graphene (graphite) and h-BN flakes to judge the crystallographic orientation, I am not certainly sure whether it is significantly better than the method established in literature (literatures, ref 20, 21 in the manuscript, also relied on the optical microscopy images to judge the crystallographic orientation). Nevertheless, I think the idea shown here is interesting.

Technically, I have few comments that would like to ask authors to consider the revision.

Comment 1) The method "using the neighboring graphite edge" is relying on two things. First is, thicker graphene (or graphite) should have edges close to principal crystallographic axes. Second is, neighboring graphite and graphene should have the same crystallographic orientation. It is not obvious to me why these two things have to be true. I would ask the author to add an explanation for this.

In particular, the second point infer that neighboring graphene and graphite used here must be exfoliated from the same single crystal grain of graphite. I wonder how the author judges that two

different graphite and graphene flakes are having the same orientation. Does all the graphene on the substrate have the same orientation under the fabrication process performed here? Or is there any rule to judge different flakes are from the same grain and having the same orientation from the optical microscope image? I think this point was not discussed in the manuscript, but may be important for the reader. I would also like the author to check this also for h-BN.

Comment 2) I am not sure whether the sentence "first to fabricate the perfect double-aligned graphene supermoiré lattice" is correct or not. I believe that their ref. [20] (<https://doi.org/10.1038/s41565-019-0547-2>) also claims similar. Please check if this statement is accurate.

Comment 3) I think it will be useful to explain in more detail about flip over methods such as condition of pickup temperature (and some other condition) of h-BN using PPC, pick up temperature (and some other condition) of h-BN from PPC to PC, and fabrication method of PPC etc.. I would also like to know the success rate of the flip over method.

Comment 4) It is interesting to see that in their double-aligned C1 ($0^\circ/0^\circ$) device, the author observed the appearance of satellite peaks at hole-side, located at $-n\pi$ and $-2n\pi$. Is it possible to explain in a little bit more detail why $-2n\pi$ peak should appear at a perfectly aligned C1 ($0^\circ/0^\circ$) device? I think this interpretation is different from ref. [20] in which the author discussed that $-2n\pi$ peak should disappear in perfectly double-aligned condition.

Comment 5) It may be useful to show Raman data to determine the relative angle between top (bottom) h-BN and graphene for device C1. It may be already shown in Fig. 3 or data in supplement, but it was not obvious from the main text how accurately determined the angle between h-BN and graphene for device C1.

Comment 6) It is a good idea to avoid overlap between x- and y-axis, and flake image in Fig. 2(a,d,g), since it is difficult to see the edge of graphite and graphene due to the axis overlapping.

Reviewer #3 (Remarks to the Author):

In this work, Hu et. al. demonstrate a stacking technique that increases the yield of hBN/graphene aligned devices. This primarily involves using standard stamping techniques but applying logic to help overcome the $1/8$ chance of aligning three layers together based on crystal edges. The three intuitions they follow are: (1) Because edges can be zig-zag or armchair, they make two parallel devices by ripping and stacking with a 30 deg rotation angle; (2) Because monolayer flakes usually don't have straight edges for visual alignment, they recognized that neighboring thicker flakes with straight edges can be used for improved alignment; (3) Because hBN layers have alternating rotation angle, they used a flip-over stacking method to retain information of the hBN crystal axes. The authors convincingly achieve the stated goals of rationally removing the $1/8$ limitation, though the stacking methods are not particularly novel. They show good characterization of the relative alignment of layers and produce some transport devices as a demonstration of the technique. The biggest deficiency in the presentation is the lack of quantitative success rate or precision, which determines how transformative the approach might be. For example, the presentation of 20 samples in the supplement is impressive, but it appears the authors haven't explicitly claimed the yield. For example, did they fabricate 20 samples, with 100% success rate, or 100 samples with 20% success rate? Also, this reviewer takes issue with the frequent use of the descriptor "perfect" and claiming zero twist angle. Since the main message of the manuscript is an improved ability to fabricate samples deterministically, it seems important to be quantitative about the precision and success rate.

There is an added consideration: namely that this method involves dividing the size of a flake by two, which when considering random bubbles/contaminants might mean it would be challenging to find a clean/useful region to make a device. The authors need to clearly define what they mean by "success" and "yield" in a way that researchers can meaningfully interpret.

Also, achieving a precision of ~ 0.2 degrees seems within the bound of what experts achieve when making twisted bilayer samples, so there isn't precisely a technical improvement on this front.

To this reviewer, this manuscript needs to demonstrate a clear improvement in device yield or fabrication time in a way that makes devices within practical reach that otherwise were not. On this front, the reviewer sees this as a potential partial success. The main new technical feat is ensuring that researchers can choose the rotational registry of the encapsulating hBN layers, which is not always pursued due to the added burden.

In its present form, this reviewer does not recommend the manuscript for publication, but believes that if the authors more quantitatively characterize the yield and that the yield is sufficiently high, this work may be suitable for Nature Communications.

Point-by-point responses to the reviewers' comments

(Our point-to-point responses are in **red**, and the corresponding changes in the manuscript are in **blue**.)

Report of Reviewer #1 (Remarks to the Author):

By controlling the rotational alignment between adjacent Van der Waals crystals, one can build a heterostructure where the two-lattice drift in and out of registry, forming a moire pattern. This additional periodicity profoundly modifies the electronic band structure. Such an effect was first realized in graphene aligned on hexagonal boron nitride (hBN), with the formation of secondary Dirac points at full-filling of the mini-Brillouin zone of the moire superlattice. However, repeatably forming such a moire superlattice of two different materials is experimentally challenging: one must precisely align two different crystals without any knowledge of the crystallographic axis. To date, this problem has been solved by rotationally aligning by eye long straight edges of exfoliated graphene/hBN which one assumes to be (approximately) well defined zigzag or armchair terminations of the crystal lattices. However, without knowledge of the explicit termination, only half of fabricated samples will form a moiré (assuming perfect alignment and perfect edges). An additional factor of two is lost if one is trying to align graphene to proximate hBN crystals in an encapsulated geometry. The authors present a clever way to mitigate these ambiguities in how the crystals should be arranged. By applying the "rip and stack" technique used to fabricate twisted bilayer graphene, two sections of the same crystal are stacked at either zero or thirty degrees relative to a straight edge of the top hBN crystal. To aid in this alignment, adjacent graphite can be used as it is likely from the same progenitor crystal. In conjunction, the authors flip adjacent hBN crystals so that they can form a doubly aligned heterostructure where both the top and bottom graphene are aligned to the proximate hBN crystals. To conclude, they present transport measurements of such a device. While I am excited by the results of this manuscript, there are substantial flaws that must be fixed before I can condone its publication. I detail my concerns below.

Response: We greatly appreciate the Reviewer for the excellent summary, positive appraisal of the work, and very constructive comments, which have motivated us to further improve our manuscript. Following the Reviewer's suggestion, we emphasize three important steps we have developed to ensure and warrant the success rate and precision of our alignment method, namely the "**Golden Rule of Three**". These steps are explained in our point-by-point responses below and explicitly described in the revised manuscript.

1. Discussions of technique, sample yield, and limitations. When discussing a new technique, it is extremely important to discuss the limitations. I found this discussion in the beginning of the manuscript very lacking. For example, when discussing the success rates for alignment of graphene/hBN heterostructures, the authors do not detail the key caveat of requiring the edges be perfectly crystalline. In graphene, it is very common to get disordered edges that are an admixture of zigzag and armchair terminations. These edges can appear straight. One keyway to rule this out is to look for multiple straight edges that are offset by integer multiples of 30 degrees.

Response: We thank the Reviewer for the helpful comments. A successful alignment of graphene/hBN relies on two things: (1) Precise identification of the principle crystallographic axes (PCA) of each layer, and (2) Correct alignment techniques as reported in our original manuscript. We agree with the Reviewer that we have mainly focussed on the second part on performing correct alignment but have not explicitly described some important caveats, for example, how to precisely identify the PCA before alignment. The Reviewer is correct; Since hBN and graphene come from two different flakes, there is a challenge to precisely identify the PCA of each flake. We have overcome this challenge and set several rules, the so-called "**Golden Rule of Three**", that have to be strictly followed to guarantee the alignment precision and success rate. The rules are as follows:

Golden Rule 1: DO NOT use the edge of graphene. Instead, use the straight edge of the neighbouring graphite. Moreover, the length of the straight edge should be no less than 100 μm (**Fig. R1a-c**).

Golden Rule 2: DO NOT use a graphite flake with a single straight edge. Instead, use a graphite flake with multiple straight edges that are offset by integer multiples of 30 degrees to each other (**Fig. R1d-f**).

Golden Rule 3: DO NOT measure the straight edge only one time. Instead, measure the angle at least three times and ensure that the error in each measurement is less than 0.2 degrees (**Fig. R1g-i**).

Below we explain why we must strictly follow the "Golden Rule of Three".

First, we do not suggest the users to use the edge of graphene, because the graphene has a poor contrast under optical microscopy, making it difficult to identify the PCA. On the contrary, the edge of thick graphite is more obvious. Moreover, in order to keep the high precision of 0.2 degrees for alignment, the error of the PCA should be controlled within 0.2 degrees. Therefore, we must choose a graphite flake with a long straight edge. The longer the edge, the smaller the error is. **Figure R2** demonstrates how the length of the graphite edge affects the angle measurement. When the length of the edge is smaller than 50 μm (**Fig. R2a-c**), the random error of PCA is larger than 0.5 degrees

even with three measurement repeats. In this case, it is impossible to realise precise alignment below 0.2 degrees. Thus, we can also explain why, in earlier reports, the misalignment is usually as large as 0.5-1 degree, as summarised in **Table S1**. On the other hand, if the length of the edge is more than 100 μm (**Fig. R2d-f**), the random error can be controlled well below 0.2 degrees. In this case, it is possible to achieve the precise alignment of below 0.2 degrees, as shown in **Fig. 2j**. **The Golden Rule #1** thus dictates that we must use graphite flakes with a straight edge of more than 100 μm (**Fig. R1a-c**).

Fig. R1: Optical images for precisely identifying the PCA. a-c, the length of graphite edge is larger than 100 μm . d-f, The angle of the graphite edge is integer multiples of 30 degrees. g-i, The error of three consecutive angle measurements $\Delta\theta$ of less than 0.2° .

Fig. R2: Optical images of angle measurement with different edge lengths. The angle error of three consecutive measurements of **a-c**, $\Delta\theta > 0.5^\circ$. **d-f**, $\Delta\theta < 0.2^\circ$.

Second, as suggested by the Reviewer, it is very common to get disordered edges that are an admixture of zigzag and armchair terminations. Therefore, we cannot identify PCA with 100% certainty if only depending on one single edge. **Our Golden Rule #2 thus dictates that we must use graphite flakes with multiple straight edges aligned at integer multiples of 30 degrees (Fig. R1d-f).**

Third, when utilising an optical microscope (in our case Nikon-LV100NDA) to identify and measure the angle of the graphite edge, the measured angles are not exact and prone to error (that can be easily larger than 0.2 degrees) due to human error and the limited optical microscope resolution. When the fluctuation of the measured angles is more than 0.2 degrees, keeping the alignment precision as good as 0.2 degrees will be impossible. **Our Golden Rule #3 thus dictates that we must identify and measure the angle of the edges at least three times and ensure the angle fluctuation is below 0.2 degrees (Fig. R1g-i).**

Strictly following the above three golden rules, we are confident that the yield of alignment can be close to 100%, as demonstrated in our manuscript. Further, the alignment precision is guaranteed to be below 0.2 degrees.

Change: On page 16, line 3, we have added the “Golden Rule of Three” in the discussion part of the revised manuscript. We also included systematic data and analysis in the supporting information Fig. S17 and Fig. S18.

2. Furthermore, the authors present a figure of merit of their technique of alignment of “~0.2 degrees.” The authors should be able to do a more rigorous assessment of their technique. Particularly, some figures of merit of alignment based on the lengths of the edges they aligned for the device. Additionally, supplementary tables S1 and S2 present the achieved angular alignment as 0 with no error bars.

Response: We thank the Reviewer for the helpful comments. We agree with the Reviewer that we should do a more rigorous assessment of our technique, and we hope the Golden Rules of Three discussed above can meet this. As the first Golden rule, also indicated by the Reviewer, we suggest using a graphite flake with a long edge larger than 100 μm .

In our work, the twist angle is measured by Raman spectroscopy, Scanning Tunneling Microscopy (STM), and electrical transport, as summarized in the method section. Thus, we agree that we need to indicate the error bar in the measured angles that can be derived from the transport. The twist angle (θ) from the moiré full filling n_s has some error because superlattice gaps are usually present over a range of n_s . Typically, we can identify the position of full filling to an accuracy of $\delta n_s \approx 10^{11} \text{ cm}^{-2}$, corresponding to a twist-angle error of $\pm 0.02^\circ$ [Nature **583**, 221 (2020)].

Change: We add this error bar in Table S1 and S2.

3. When discussing possible configurations, there are eight possible configurations (assuming perfect alignment), as the top hBN can be either armchair or zigzag in its termination (there are actually even more when one considers the c-axis termination of the hBN). This is incorrect in both the main text and figure 1.

Response: We thank the Reviewer for the helpful comments and apologise for the unclear description of Fig. 1. We completely agree with the Reviewer that there are eight possible configurations when considering the c-axis of hBN; This is the reason why we can remove the 1/8 (12.5%) possibility limitation by using our technique. As we describe in the main text: “Moreover, the lattice symmetry of the hBN layer also obscures the inversion symmetry of the moiré heterostructure, further decreasing the success rate to 12.5%”. We also consider the c-axis termination of hBN in Fig. 3. As

we describe in the main text: "Depending on the surface symmetry determined by the layer number of hBN, the final stack can be $0^\circ/0^\circ$ or $0^\circ/60^\circ$ (Fig. 3b, c)."

We apologise for the misunderstanding. For simplicity, we initially did not include the c-axis termination consideration and thus, only four possible configurations were originally drawn in Fig. 1. To avoid misunderstanding, we have now revised Fig.1 and included all eight possible configurations. When the top and bottom hBN have the same symmetry, for example, B, N, B, N (Top) – B, N, B, N (Bottom), we use the "C" to represent the configuration, and there are 4 configurations. When the top and bottom hBN has different symmetry, for example, B, N, B, N (Top) – N, B, N, B (Bottom), this corresponds to a rotation of 60° between G and B-hBN. In this case, we use the "C*" to represent the configuration, and there are again 4 configurations. Now, there are eight possible configurations in total, as shown in Fig. R3.

Fig. R3: Eight possible configurations considering both the edge chirality and crystal symmetry of graphene and hBN.

Change: We have revised Fig.1 in the main text accordingly. On page 6, line 7, we add: "When in the assembly of hBN/graphene/hBN sandwich structure, there are eight possible configurations: C1 ($0^\circ/0^\circ$) & C1* ($0^\circ/60^\circ$), C2 ($0^\circ/30^\circ$) & C2* ($0^\circ/90^\circ$), C3 ($30^\circ/30^\circ$) & C3* ($30^\circ/90^\circ$) and C4 ($30^\circ/0^\circ$) & C4* ($30^\circ/60^\circ$) (Fig. 1a). All possible configurations lead to 1/2 (50%) success rate for single alignment (C2, C2*, C4, C4*), and 1/8 (12.5%) success rate for double alignment (C1)."

4. Using adjacent graphite to align graphene is a clever technique and I applaud the authors in making many well aligned devices using this technique. However, the authors do not discuss any limitations, which are particularly interesting. Figure 2 shows that they are already succeeding in using graphite ~40 microns away to align graphene. Did any samples fail using this technique not come out as intended? What is the furthest away piece of graphite they have used for successful alignment? It certainly will not always be the case that adjacent graphene and graphite will share the same crystallographic axis.

Response: We thank the Reviewer for these interesting questions. The prerequisite to using the adjacent graphite is that the adjacent graphene and graphite should share the same PCA. This depends on how we exfoliate graphene, as shown in **Fig. R4**.

Generally, there are two kinds of exfoliation methods: one is the overlap exfoliation (**Fig. R4a-c**). Please see this video as a reference:

https://www.google.com/search?q=how+to+exfoliate+graphene&newwindow=1&sxsrf=AJOqlzWp_eRHAPG6dNGNFvYPVda94Fugw:1678597987485&source=Inms&tbm=vid&sa=X&ved=2ahUKEwiR547mONX9AhUdlbcAHRMoDLYQ_AUoAXoECAEQAw&biw=1707&bih=916&dpr=1.5#fpstate=ive&vld=cid:e40e9a01,vid:wa0020125sU

In this case, as indicated by the Reviewer, it certainly will not always be the case that adjacent graphene and graphite will share the same PCA because two adjacent grains can be at any random orientation (**Fig. R4b, c**). On the contrary, we use the so-called "non-overlap exfoliation" method, as depicted in **Fig. R4d-f**. Here, we carefully exfoliate the graphene flakes multiple times and make sure the two domain flakes are large and do not overlap each other. In this case, in principle, their PCA should be the same within a radius of $R \approx 250 \mu\text{m}$ (**Fig. R4e**). It should be pointed out that with optimization of the exfoliation process, the radius R can be even as large as 300 or 400 μm , depending on the size of the exfoliated single grain. As a result, graphene and neighbouring graphite can have a high chance of sharing the same PCA.

Fig. R4. Two methods of exfoliation of graphene flakes. a-c, Overlap exfoliation. d-f, Non-overlap exfoliation.

Fig. R5. The influence of the distance between graphene and graphite on the alignment. a-c, Optical images of graphene flakes with a distance of 40, 130 and 250 μm . d-f, The graphene/hBN stack using the 30° rotation technique. g-i, Typical Raman data from d-f.

Regarding the distance between graphene and graphite, we have studied three different cases with a distance of 40, 130 as well as 250 μm as shown in **Fig. R5**. We found that in all three cases, the neighboring graphite edge can be used for alignment, as the Raman 2D peak shows that they are all close to 41 cm^{-1} . Therefore, we strongly recommend that potential users utilize our exfoliation method to ensure that the adjacent graphene and graphite share the same PCA.

Change: On page 9, line 14, we add: "Regarding the distance between graphene and graphite, we have studied three different cases with a distance of 40, 130 as well as 250 μm (**Fig. S8**). We found that in all three cases, the neighbouring graphite edge can be used for alignment if we use the so-called "non-overlap exfoliation" method (**Fig. S9**)."

5. hBN flipping. The discussion motivating the flipping of hBN is lacking. The authors do not mention that the hBN grown by NIMS is AA' stacking. Therefore, motivating the idea of trying to control the proximate hBN surface with the graphene by such a flipping technique. However, this appears to not be controlled at all as the two sections of hBN used in both the presented devices appear to be different colors (as seen by a crude check of their RGB values. even if they seemed to be the same color, I would not trust them to have the exact same number of layers as they are disconnected). There is then no control of inversion symmetry within the stacks.

Response: We thank the Reviewer for the critical comments. Here, we would like to explain the principle of the flip-over technique, as shown in **Fig. R6**. In the conventional pick-up technique, we pick up BN1 and place it on top of BN2 (with a graphene layer in between). In this case, the bottom face of BN1 will be in proximity to the top face of BN2. Thus, the two adjunction surfaces do not come from the same BN layer position (i.e., one is from the bottom and the other from the top). To have the two adjunction surfaces coming from the same position (for example from the bottom face), we flip BN2 and then place BN1 on top of BN2. This is called the flip-over technique.

Further, we can consider three different routes for obtaining BN1 and BN2 to be used during the flip-over technique. First, hBN can be cut into two (BN1 and BN2). In this case, we can make sure that BN1 and BN2 not only share the same PCA, but also the same surface (**Fig. R6 b,c**). However, this cutting process is tedious, requiring complicated lithography steps. Second, BN1 and BN2 can come from naturally fractured hBN flakes, which can always be found during mechanical exfoliation, as shown in **Fig. R6 d-e**. Third, we can also obtain BN1 and BN2 from two neighbouring hBN flakes whose PCA have an integer multiple of 30 degrees to each other, as shown in **Fig. R7**. The same as

graphene, hBN also has a hexagonal symmetry. If the edge angle of the adjacent hBN is an integer multiple of 30° , then the two adjacent hBN flakes have a high chance of coming from the same crystal. This is because if they do not come from the same crystal, the probability of its edge oriented at an integer multiple of 30 degrees ($n \times 30^\circ$, $n = 0, 1, 2 \dots 11$) is quite low, i.e., $12/360$ (3.3%). Therefore, in this case, the two hBN flakes can also be used for alignment, as we demonstrated in Fig. 3f,g.

Fig.R6. The flip-over technique. a, Illustration of conventional pick-up technique and flip-over technique. b,c, Optical images of two hBN crystals etched with DRIE into two isolated parts. d,e, Optical images of two fractured hBN after mechanical exfoliation.

Fig. R7. The adjacent hBN flakes share the same PCA and the same surface. a, Illustration of the relationship between angles and the chirality of the adjacent edges, adapted from [APL. 93, 163112 (2008)]. Optical images of adjacent hBN flakes with edge angles of b, 0 degrees and c, 60 degrees.

Change: On page 12, line 3, we add: “Based on this concept, we can consider three different routes for obtaining BN1 and BN2 to be used in the flip-over technique (Fig. S14).”

6. Transport measurements. In discussing the Landau fans, it would be nice to have some comparison to other doubly aligned devices, as the authors claim they have achieved some of the best alignment. For example, Fig. 4F and Fig. 4G show fans emerging from integer number of chair carriers not equal to 4 or 8. These features can be ascribed to a simple Hofstadter picture, but I am not sure if some of them have been so cleanly observed before.

Response: We thank the Reviewer for the questions. Yes, we have shown another doubly aligned device in supporting information **Fig. S16**. Consistent with the first device (**Fig. 4f**), we observed a similar Hofstadter picture in this second device, further supporting the reliability of our reported technique. Moreover, we also performed an analysis of the Hofstadter butterfly and Brown-Zak oscillations of the second device using the Wannier diagram, as shown in **Fig. S16**.

We also thank the Reviewer for noticing the anomalous feature in the Landau fan diagram (**Fig. 4f and Fig. 4g**), where the trajectory of quantum oscillations looks not equal 4 or 8. Actually, similar phenomena were previously observed in single-aligned devices by transport study [Nature Communications, 11, 5756 (2020)] as well as by optical study [Sci. Adv. 2:e1600002 (2016)], as shown in **Fig. R8**. In the transport study, an unusual feature appears around -42 V (second Dirac point), highlighted by the red rectangle box (**Fig. R8b**). While in the optical study, a photo-Nernst current is drastically enhanced near -4.8 V (second Dirac point), also highlighted by the red rectangle box (**Fig. R8c**). Both these two features are close to the second Dirac point.

Generally, these unusual features can be attributed to the low-energy Van Hove singularities. In the graphene/hBN moiré superlattice electronic band (**Fig. R8a**), the Van Hove singularities arise from saddle point formation in the moiré minibands, indicated by the solid white arrows in **Fig. R8a**. These band singularities feature Lifshitz transitions, where the number of Fermi contours will change, leading to a large carrier density. When the Landau level crosses these singularities, then the shape of the Landau fan will be deformed, and it will finally lead to an anomalous feature, as shown in **Fig. R8b, c**.

However, we should note that these observations were only limited to the single-aligned device. We are the first to observe these unusual features in the double-aligned device. Currently, we are

carefully working on these phenomena, and we believe lots of interesting physics are waiting for us to discover in this novel system.

[redacted]

Fig. R8. Anomalies induced by Lifshitz transitions. **a**, VHSs in graphene/hBN moiré minibands. **b**, Landau fan diagram of single-aligned graphene/hBN device. **c**, Longitudinal resistance R_{xx} and photocurrent generation as a function of gate voltages under a magnetic field. (a, c) adapted from [Sci. Adv. 2:e1600002 (2016)], (b) adapted from [Nature Communications 11, 5756 (2020)].

Change: On page 15, line 17, we add: “Here, we note that apart from the integer filling fractions of 0, -4 , -8 in the Landau fan diagram, there are also some anomalous features between them, which can be ascribed to the low energy Van Hove singularities. Similar phenomena were previously observed in single-aligned devices by transport study⁴⁰ as well as the optical study⁴¹.”

7. Miscellaneous points. When plotting Raman or transport data, do not use jet. It is misleading in that it provides substantial contrast to the eye where there should not be. It is technically most closely a divergent colormap and therefore not an appropriate choice for a strictly positive number anyway. Please use a perceptually uniform sequential colormap.

(<https://matplotlib.org/stable/tutorials/colors/colormaps.html>, this link also has interesting discussion on why such choices are important).

Response: We thank the Reviewer for the helpful comments. We have carefully studied the link shared by the Reviewer, and we have learned a lot about how to choose the colour map. For our Raman map data, since we only have two discrete values of 40 cm^{-1} for 0° G/hBN and 20 cm^{-1} for 30° G/hBN , both jet and uniform colour maps can provide the contrast. However, since our transport data show continuous resistance values, we agree with the Reviewer that a uniform sequential colour map is better. Below, we compare the jet colourmap (Fig. R9a-c) and the uniform sequential colourmap (Fig. R9d-f). In the main text, we have changed the Landau fan diagram into a uniform sequential colour map.

Fig. R9. Landau fan diagrams using **a-c**, the jet and **d-f**, the uniform sequential colourmap.

Change: We have revised the manuscript and used a uniform sequential colour map in Fig. 4f, g, h.

8. In the supplemental materials, while it is discussed elsewhere, the methods section does not detail the hBN flipping technique.

Response: We thank the Reviewer for the helpful comments. Following the Reviewer's suggestion, we add the details of the hBN flipping technique in the methods section, including the temperature and the detail of the PC and PPC we use. This information is also requested by Reviewer #2.

Change: In the method section, we add: "The flip-over technique includes three steps. First, we used a thin film of polycarbonate (PC, Sigma Aldrich, 6% dissolved in chloroform purchased at HQ Graphene) and polydimethyl-siloxane (PDMS) stack on a glass slide to pick up the first piece of hexagonal boron nitride flake (BN1) at 60 °C. Then we used the van der Waals force between BN1 and monolayer graphene to tear and pick up half of the graphene flake. The remaining graphene flake on the silicon was rotated by 30° and picked up at 40 °C. Second, the polypropylene carbonate (PPC) layer was spun on top of a bare silicon wafer, released with a hollow-shaped Scotch tape, and then transferred on top of the second PDMS stack on a glass slide. The PPC (Sigma-Aldrich, CAS 25511-85-7) is made of propylene carbonate and anisole with a mass fraction of 5%. To enhance the interaction between PPC and PDMS, the PDMS is treated with oxygen plasma for 10 mins before being covered with PPC film. Then the PPC/PDMS stamp is used to pick up the second piece of hexagonal boron nitride flake (BN2) at 60 °C. Third, in order to expose the bottom surface of BN2, we flip over the PDMS/PPC/BN2 and make the bottom surface of BN2 to be exposed to the bottom surface of BN1. Finally, we use PC/BN1/G to pick up BN2 from PPC at 80-100 °C. The success of this procedure relies on the stronger adhesion between PC and BN1 compared to BN2 and PPC."

Report of Reviewer #2 (Remarks to the Author):

In this article, the author presented a strategy to construct a double-aligned hBN/graphene/hBN moiré superlattice where crystallographic orientation of both top and bottom h-BN is aligned to that of graphene located in between h-BN. A fabrication of double-aligned hBN/graphene/hBN is recognized as a challenging subject in the field of 2D materials, primarily due to the uncertainty of the edge chirality of graphene and h-BN. In my understanding, the author provided three experimental techniques to improve the reliability for fabricating double-aligned hBN/graphene/hBN. These are,

- 1) Use 30-degree rotated graphene flakes.
- 2) Using neighboring graphite edges to determine principal crystallographic axes of graphene.
- 3) Flip-over technique to control the alignment of top hBN and bottom hBN.

By using these techniques, the author achieved the fabrication of the following device.

- 4) Perfect double-aligned hBN/graphene/hBN moire superlattice.

These 1) to 4) can be regarded as a novelty of this manuscript. The method 1), 2), and 3) is a good idea and I think these ideas were not reported before. By using method 1), 2), and 3), the author improved the fabrication yield of double-aligned hBN/graphene/hBN devices. This may be a good improvement. Since this method still relied on the optical microscopy image of the graphene (graphite) and h-BN flakes to judge the crystallographic orientation, I am not certainly sure whether it is significantly better than the method established in literature (literatures, ref 20, 21 in the manuscript, also relied on the optical microscopy images to judge the crystallographic orientation). Nevertheless, I think the idea shown here is interesting. Technically, I have few comments that would like to ask authors to consider the revision.

Response: We greatly appreciate the Reviewer for the excellent summary of our work and for recognising the novelty of this manuscript. Especially the three main methods of our alignment technique were correctly summarised by the Reviewer. We also thank the Reviewer's suggestions and comments, which have motivated us to further improve our manuscript. Before we address them individually, we would like to clarify the advantages and significance of our technique compared to those reported earlier [20, 21].

References [20 and 21] are the two earlier efforts that tried to control the alignment by dynamically rotating the top hBN (illustrated in Fig. R10a), while monitoring the change in the properties of the system. Therefore, there is no need to use optical microscopy images to judge the PCA in their works.

Instead, the methods produce a certain unknown twist-angle, and determining this resulting twist-angle will only depend on in-situ Raman measurement. As a result, the methods have no control over the exact alignment of the device and does not allow fixing the alignment at a predefined angle. Thus, we would say that our static method (illustrated in Fig. R10b) is preferred and will be appreciated by the 2D community. In particular, our technique focuses on zero-degree alignment with high precision to produce “perfect” double-aligned samples.

Fig. R10. Comparison of our work with previous works. Illustration of **a**, Rotation alignment, **b**, Optical edge alignment.

1. The method “using the neighboring graphite edge” relies on two things. First is, thicker graphene (or graphite) should have edges close to principal crystallographic axes. Second is, neighboring graphite and graphene should have the same crystallographic orientation. It is not obvious to me why these two things have to be true. I would ask the author to add an explanation for this.

Response: We thank the Reviewer for the comments. Yes, when using the neighbouring graphite edge, the two things indicated by the Reviewer have to be true. As we also reply to Question #4 of Reviewer#1, this depends on how you exfoliate graphene, as shown in Fig. R11.

Generally, there are two kinds of exfoliation methods. One is the “overlap exfoliation” method (Fig. R11a-c). In this case, as indicated by the Reviewer, it certainly will not always be the case that adjacent graphene and graphite will share the same PCA because two adjacent crystals can be at

any random orientation (Fig. R11b, c). However, we use another method called "non-overlap exfoliation", as shown in Fig. R11d-f. In this method, we carefully exfoliate the graphene flakes multiple times and ensure the two graphite crystals are large and do not overlap each other. In this case, in principle, within a radius R , their PCA should be the same (Fig. R11e). To achieve the goals described in our manuscript, we thus suggest potential users use our exfoliation method. For a more detailed explanation, we refer the Reviewer to question 4 of Reviewer #1.

Fig. R11. Two methods of exfoliation of graphene flakes. a-c, Overlap exfoliation. d-f, Non-overlap exfoliation.

Change: On page 9, we add: "Regarding the distance between graphene and graphite, we have studied three different cases with a distance of 40, 130 as well as 250 μm (Fig. S8). We found that in all three cases, the neighboring graphite edge can be used for alignment if we use the so-called "non-overlap exfoliation" method (Fig. S9)."

2. In particular, the second point infers that neighboring graphene and graphite used here must be exfoliated from the same single crystal grain of graphite. I wonder how the author judges that two different graphite and graphene flakes have the same orientation. Does all the graphene on the substrate have the same orientation under the fabrication process performed here? Or is there any rule to judge different flakes are from the same grain and having the same orientation from the

optical microscope image? I think this point was not discussed in the manuscript but may be important for the reader. I would also like the author to check this also for h-BN.

Response: We thank the Reviewer for the comments. Yes, there is a rule that the angle between straight edges of graphite and graphene should be an integer multiple of 30 degrees, as illustrated in Fig. R12 b,c. Because both Graphene and hBN flakes have hexagonal symmetries, when the angle of the adjacent edge is an integer multiple of 30° , these two flakes usually come from the same crystal. This is because if they do not come from the same crystal, the probability of its edge oriented happen at an integer multiple of 30 degrees ($n \times 30^\circ$, $n = 0, 1, 2 \dots 12$) is quite low, i.e., $12/360$ (3.3%). In most probability, they come from the same crystal.

[redacted]

Fig. R12. The adjacent graphene flakes share the same PCA. a, Illustration of the relationship between angles and the chirality of the adjacent edges, adapted from [APL 93, 163112 (2008)]. Optical images of adjacent graphene flakes with an angle of b, 30 degrees and c, 90 degrees.

Change: On page 16, line 3, we have added the “Golden Rule of Three” in the discussion part of the main revised manuscript. We also included systematic data and analysis in the supporting information Fig. S17 and Fig. S18.

3. I am not sure whether the sentence “first to fabricate the perfect double-aligned graphene supermoiré lattice” is correct or not. I believe that their ref. [20] (<https://doi.org/10.1038/s41565-019-0547-2>) also claims similar. Please check if this statement is accurate.

Response: We thank the Reviewer for the comments. We indeed also noticed that Refs. [20, 21] seemed to indicate achieving a perfect double-aligned graphene supermoiré lattice. However, there

was no direct proof indicating a perfect alignment. Reference [20] is particularly interesting since they showed how electrical transport and Raman data dynamically change with twist angles, but the focus of the work was not on a perfect alignment.

As indicated by the Reviewer (see also Question #5), ref. [20] only shows $-n_s$ peak, whereas $-2n_s$ is missing in their devices. This is possibly due to the special rotating device, making it difficult to apply a higher gate voltage equivalent to $-2n_s$. On the other hand, our work focuses on zero-degree-aligned devices with improved alignment precision. We combine Raman, STM and low-temperature transport to clearly show the perfect double-aligned structure. The transport data also agree with our theoretical calculation. Because this discussion also relates to Question #5, we refer the Reviewer to our answer to Question #5 below for more details.

4. I think it will be useful to explain in more detail about flip over methods such as condition of pickup temperature (and some other condition) of h-BN using PPC, pick up temperature (and some other condition) of h-BN from PPC to PC, and fabrication method of PPC etc. I would also like to know the success rate of the flip over method.

Response: We thank you for the Reviewer's helpful comments. We have added the details about the flip-over technique in the method section. If users can strictly follow the "Golden Rule of Three" (please see our answer to Question #1 of Reviewer #1) as well as the procedures below, we can confidently say that the success rate is close to the theoretical value of 100%. The gap between experimental value and theoretical value is due to solely human errors since the transfer is highly dependent on personal experience and operation.

Change: In the method section, we add: "The flip-over technique includes three steps. First, we used a thin film of polycarbonate (PC, Sigma Aldrich, 6% dissolved in chloroform purchased at HQ Graphene) and polydimethyl-siloxane (PDMS) stack on a glass slide to pick up the first piece of hexagonal boron nitride flake (BN1) at 60 °C. Then we used the van der Waals force between BN1 and monolayer graphene to tear and pick up half of the graphene flake. The remaining graphene flake on the silicon was rotated by 30° and picked up at 40 °C. Second, the polypropylene carbonate (PPC) layer was spun on top of a bare silicon wafer, released with a hollow-shaped Scotch tape, and then transferred on top of the second PDMS stack on a glass slide. The PPC (Sigma-Aldrich, CAS 25511-85-7) is made of propylene carbonate and anisole with a mass fraction of 5%. To enhance

the interaction between PPC and PDMS, the PDMS is treated with oxygen plasma for 10 mins before being covered with PPC film. Then the PPC/PDMS stamp is used to pick up the second piece of hexagonal boron nitride flake (BN2) at 60 °C. Third, in order to expose the bottom surface of BN2, we flip over the PDMS/PPC/BN2 and make the bottom surface of BN2 to be exposed to the bottom surface of BN1. Finally, we use PC/BN1/G to pick up BN2 from PPC at 80-100 °C. The success of this procedure relies on the stronger adhesion between PC and BN1 compared to BN2 and PPC."

5. It is interesting to see that in their double-aligned C1 ($0^\circ/0^\circ$) device, the author observed the appearance of satellite peaks at hole-side, located at $-n_s$ and $-2n_s$. Is it possible to explain in a little bit more detail why $-2n_s$ peak should appear at a perfectly aligned C1 ($0^\circ/0^\circ$) device? I think this interpretation is different from ref. [20] in which the author discussed that $-2n_s$ peak should disappear in perfectly double-aligned condition.

Response: We thank the Reviewer for the question. In fact, this question helps us reveal a unique property of the double-aligned system. In stark contrast to the previous single-aligned devices, where only $-n_s$ state can be observed [Nature 497, 594–597 (2013); Nature 497, 598–602 (2013)], the increase of the bandgap and induced band isolation because of the double moiré make it possible for us to observe the $-2n_s$ peak.

The emergence of the $-2n_s$ peak can be understood by our band structure calculation (Fig. R13a). It can be seen that apart from the gap between V1 and V2, which leads to the $-n_s$ peak, there is also a significant band gap located between V2 and V3. When the Fermi level is tuned and moved into this band gap, the $-2n_s$ peak can be observed in R_{xx} , while the sign of R_{xy} is reversed (Fig. R13b).

As explained in our answer to Question #3, Ref. [20] only produces the $-n_s$ peak because of the limitation in the applied gate voltage related to the special device design. If a perfect double alignment is indeed realised, the $-2n_s$ peak is expected to emerge at a higher gate voltage, as observed in our devices but missing in Ref [20]. We hope our transport data of the perfect double-aligned device can be used as a reference for future studies.

Fig. R13. Band structure and transport data of the double-aligned device. a, Band structure of a perfectly double-aligned system. **b,** R_{xx} and R_{xy} change with carrier density at 2 K.

6. It may be useful to show Raman data to determine the relative angle between top (bottom) h-BN and graphene for device C1. It may be already shown in Fig. 3 or data in supplement, but it was not obvious from the main text how accurately determined the angle between h-BN and graphene for device C1.

Response: We thank the Reviewer for the helpful comments. The angle of device C1 is determined by both Raman and transport data. The detail on how to determine this angle is shown in the method section. Generally, as the Reviewer indicated, we first use Raman to verify the sample twist angle and a typical Raman data of a perfectly double-aligned device is shown in Fig. 3. Subsequently, a standard lithographic technique is used to pattern a Hall bar geometry.

Change: On page 14, line 1, we add: “We first use Raman to verify the sample twist angle, and typical Raman data of a perfectly double-aligned device is shown in Fig. 3. Subsequently, a standard lithographic technique is used to pattern a Hall bar geometry.”

7. It is a good idea to avoid overlap between x- and y-axis, and flake image in Fig. 2(a,d,g), since it is difficult to see the edge of graphite and graphene due to the axis overlapping.

Response: We thank the Reviewer for this careful observation. Following Reviewer's suggestion, we slightly shift the x- and y-axis and now the edge of graphite and graphene can be clearly seen.

Fig. R14. Optical images of graphene and graphite flakes.

Change: We changes Fig. R14 into Fig.2.

Report of Reviewer #3 (Remarks to the Author):

In this work, Hu et. al. demonstrates a stacking technique that increases the yield of hBN/graphene aligned devices. This primarily involves using standard stamping techniques but applying logic to help overcome the 1/8 chance of aligning three layers together based on crystal edges. The three intuitions they follow are: (1) Because edges can be zigzag or armchair, they make two parallel devices by ripping and stacking with a 30 deg rotation angle; (2) Because monolayer flakes usually don't have straight edges for visual alignment, they recognized that neighboring thicker flakes with straight edges can be used for improved alignment; (3) Because hBN layers have alternating rotation angle, they used a flip-over stacking method to retain information of the hBN crystal axes.

The authors convincingly achieve the stated goals of rationally removing the 1/8 limitation, though the stacking methods are not particularly novel. They show good characterization of the relative alignment of layers and produce some transport devices as a demonstration of the technique.

Response: We appreciate the Reviewer for the excellent summary and positive appraisal of our work. We agree with the Reviewer that the basis of our stacking method is not particularly novel since it is based on traditional PDMS stamping and requires no specialized or complicated equipment. The three points of intuition summarised by the Reviewer indeed describe the logical thinking of our method, which is the strength of our method. That is why we are confident that our technique can be quickly learned and followed by the 2D community to help the research of "Twistronics" move forward. The novelty of our manuscript is in the control alignment, for which three main techniques are involved and critical. As also pointed out by Reviewer#2, these ideas were not reported before. Below we address the Reviewer's comments point by point.

1. The biggest deficiency in the presentation is the lack of quantitative success rate or precision, which determines how transformative the approach might be. For example, the presentation of 20 samples in the supplement is impressive, but it appears the authors haven't explicitly claimed the yield. For example, did they fabricate 20 samples, with 100% success rate, or 100 samples with 20% success rate? Also, this reviewer takes issue with the frequent use of the descriptor "perfect" and claiming zero twist angle. Since the main message of the manuscript is an improved ability to fabricate samples deterministically, it seems important to be quantitative about the precision and success rate.

Response: We thank the Reviewer for the critical comments. We understand the Reviewer's main concern regarding the quantitative success rate and precision of our technique. We have been working on this technique for nearly 3 years. We solved various technical problems through logical thinking, finally realised the high yield, controlled alignment and removed the 1/8 limitation in the double-aligned hBN/Graphene/hBN heterostructure. Moreover, we also show that our technique can be extended to the strongly correlated systems, as demonstrated by the observation of the "correlated insulating states" in aligned ABC-staked trilayer graphene/hBN (Fig. S20). Frankly, we have mastered this technique in such a way that it has become routine for us to produce double-aligned structures. As long as our "Golden Rule of Three" is strictly followed, as described in our answer to Reviewer #1, we can confidently say that the actual (theory) success rate should be 100%, but there is always a margin of error due to human nature (Statistically, we will have 2-3 failures out of 30 tries, and thus our success rate is >90%). Any failures are mainly due to human errors, as the transfer is highly dependent on personal experience and operation. In comparison, the success rate demonstrated by an earlier work [Nat Commun 12, 7196 (2021)] was only ~8% (5 successful samples out of 63 devices) using the traditional random stacking method, which is close to the theoretical value of 1/8 (12.5%). Our technique thus demonstrates significant progress in this field, as also summarised in Table S1 and Table S2.

2. There is an added consideration: namely that this method involves dividing the size of a flake by two, which when considering random bubbles/contaminants might mean it would be challenging to find a clean/useful region to make a device. The authors need to clearly define what they mean by "success" and "yield" in a way that researchers can meaningfully interpret.

Response: We thank the Reviewer for the helpful comments. We define our work as "success" as it can remove the 1/8 (12.5%), allowing a "alignment yield" of close to 100% to be realised for the double-aligned hBN/graphene/hBN heterostructure. Moreover, the twist angle between each layer can be controlled well below 0.2 degrees.

However, we also agree with the Reviewer that for a successful method to be considered, it should be able to produce high-quality devices (fewer damages, bubbles, contaminants, etc.). With regard to these issues, we summarise various possible problems during device fabrications and our

strategies to minimise such problems (Table. R1 and Fig. R15). But, in this manuscript, we only focus on the alignment problem.

Problems	Reasons	Strategies	References
Alignment	Uncertainty in the edge chirality and crystal symmetry ; error in identifying the PCA	Rotation 30° technique; Flip-over technique; Using neighboring graphite edge	Our work
Bubble	Contaminants; O ₂ and H ₂ O; air trap; Non-uniform hBN surface	Hot pick-up technique	Nat Commun 7 , 11894 (2016). Nat Commun 9 , 5387 (2018).
Wrinkle	Strain on hBN; Soft PDMS ; Temperature too high	Hard PDMS; Thick hBN; Moderate temperature	Our experience
Damage	Stronger interaction between graphene and silicon wafer	Less time of oxygen-plasma treated silicon wafer	Our experience

Table R1: Summary of the four main problems during transfer.

Fig.R15. Optical images of hBN/G/hBN stack. **a-c**, Standard stacks without any bubble and damage. **d-f**, Stacks with lots of wrinkles and damages. **g-i**, Stacks with lots of bubbles. A clean graphene area is highlighted by the white dashed line in each stack.

Figures R15a-c show our standard hBN/G/hBN stacks, and more than 90% of our hBN/G/hBN stacks are free of bubbles/damages/wrinkles, as demonstrated in Figures R15a-c. As can be seen from the figure, there is no bubble, wrinkle (Wrinkles will lead to the inhomogeneity of the moiré pattern) and damage within the graphene area highlighted by the white dashed lines. These successes come from our many years of experience and hard work. However, sometimes we also meet the problems of having damage, wrinkle (**Fig. R15d-f**), or bubbles (**Fig. R15g-i**) in our samples. We note that such problems were common in the early stage of this research, especially those performed by beginners. Regarding the bubble problems referred by the reviewers, they have been well addressed in previous reports (**See table R1**). Even though it is difficult to absolutely eliminate these bubbles, an earlier study also showed that the formation of bubbles is a self-cleansing process [Nano Lett. 14, 6, 3270 (2014)], which takes place at the interfaces between graphene and hBN. The surface contamination assembles into large pockets allowing the rest of the interface to become atomically clean. As shown in **Fig. R15g-i**, we can always find these atomically clean areas, which may be only 10-20 μm , but this size is large enough for us to make a Hall bar structure.

Therefore, we would like to clarify that the problems of bubbles, damage and wrinkles are not a serious issue in our study because we rely on our own experience (**Summarized in Table R1**) to effectively minimize these problems. Instead, in this manuscript, we only focus on the alignment problem, the first problem among others summarized in **Table R1**. We can increase the double alignment yield from a theoretical limitation of 12.5% to close to 100% now. As for the wrinkle and damage problems, these would be a great focus of future work.

3. Also, achieving a precision of ~ 0.2 degrees seem within the bound of what experts achieve when making twisted bilayer samples, so there isn't precisely a technical improvement on this front.

Response: We thank the Reviewer for the critical comments. Respectfully, we believe the Reviewer has mixed between twisted bilayer graphene work (G/G) and our work (G/hBN). Below, we explain the differences and the challenges.

The main problem with the G/hBN alignment originates from the fact that graphene and hBN come from two different flakes and different silicon wafers (**Fig. R16a**). To perform the alignment, the first challenge is how to precisely identify the PCA on their respective wafers. For this, we develop the “Golden Rule of Three” to guide users in identifying the PCA accurately. For a more detailed explanation, we refer the Reviewer to question 1 of Reviewer #1.

Fig. R16. Comparison of alignment of graphene/hBN and graphene/graphene. a, PCA of graphene and hBN. **b,** Alignment of graphene and hBN. **c,** PCA of graphene and graphene. **d,** Alignment of graphene and graphene.

On the other hand, for the G/G case (twisted bilayer graphene), there is no need to identify the PCA and consider the edge uncertainty because the two pieces of graphene come from the same flake (**Fig. R16c**) and the 0.2° error indicated by the Reviewer only comes from the lattice relaxation between two graphene sheets and the precision of the rotation module. Usually, the motorised in-plane rotation module and controller, applied to directional twisted bilayer graphene, should have a minimum rotation accuracy of 0.01° . If the rotation accuracy is 0.1° , it is impossible to control within 0.2° . We summarise the difference between the two systems below:

System	Magic angle (Degree)	Moiré pattern (nm)	Identify the PCA	Edge uncertainty	Energy stable	Alignment (0 Degree)	Physics
G/hBN	0	14	Yes	Yes	Yes	Challenging to control	Hofstadter Butterfly
G/G	1.1	13.3	No need	No	No	Can be well controlled	Correlated states

Table. R2. The difference between G/hBN and G/G during alignment.

The technical improvement in our work is that we can realise the <0.2 -degree precision for the alignment of graphene and hBN. While in previous studies, the precision is only 0.5-1 degrees, as we summarised in **Table S1**. We emphasise that the alignment procedure for graphene and hBN (ours) is not the same and is at a different level compared to the alignment for the twisted bilayer graphene [Nano Lett. 16, 3, 1989 (2016)].

4. To this reviewer, this manuscript needs to demonstrate a clear improvement in device yield or fabrication time in a way that makes devices within practical reach that otherwise were not. On this front, the reviewer sees this as a potential partial success. The main new technical feat is ensuring that researchers can choose the rotational registry of the encapsulating hBN layers, which is not always pursued due to the added burden. In its present form, this reviewer does not recommend the manuscript for publication, but believes that if the authors more quantitatively characterize the yield and that the yield is sufficiently high, this work may be suitable for Nature Communications.

Response: We thank the Reviewer for the helpful comments. As indicated by the Reviewer, the main technical feat is ensuring that researchers can choose the rotational registry of the encapsulating hBN layers, which is not always pursued due to the uncertainty in edge and lattice symmetry. This is

exactly what we can achieve using the technique proposed in this manuscript. Moreover, our technique can greatly improve the efficiency of making samples. Below, we would like to emphasise the improvement in alignment yield, fabrication time and precision offered by our technique.

Alignment yield. The success rate demonstrated by an earlier work [Nat Commun 12, 7196 (2021)] using the traditional random stacking method was only ~8% (5 successful samples out of 63 devices), which is close to the theoretical limitation of 1/8 (12.5%). As long as our “Golden Rule of Three” is strictly followed, we can confidently say that the success rate can be more than 90%, which is close to the theoretical value of 100%. The gap between experimental value and theoretical value can be made by the human errors since the transfer is highly dependent on personal experience and operation.

Precision. Using a traditional method (Table S1), the achieved precision is typically around 0.5-1 degree. One typical example can be seen in our previous work [Nat. Nanotechnol. 17, 378–383 (2022)], where the twist angle between graphene and hBN is as large as 0.85 degrees. Using our alignment technique and strictly following the “Golden Rule of Three” guarantee a precision of better than 0.2 degrees.

Fabrication time. Traditionally, finding a single layer of graphene with a straight edge for the alignment process is time-consuming. Our technique proposes that neighbouring graphite can also be used for alignment. Timewise, we can confidently say our technique is 10 times more efficient on average compared to other traditional methods. We summarise the improvement below:

Technical indicators	Traditional technique	Our technique
Alignment yield	8% (12.5%)	>90% (100%)
Precision	0.5-1 degree	<0.2 degree
Fabrication time	10-20 hours/1 sample	1-2 hours/ 1 sample

Table R3. Quantitatively characterizing the alignment yield, precision, and fabrication time of our technique.

From the table above, we can see that there is a clear improvement in device yield, precision and fabrication time using our technique. We hope that with the help of our technique, the fabrication of moiré samples will become easier from now on. I hope the reviewers feel the same way as we do.

Change: We have added the related discussion in the summary of Table S3.

REVIEWER COMMENTS

Reviewer #1 (Remarks to the Author):

The authors have made significant revisions to their manuscript. Overall, it is a more solid manuscript and I still believe the manuscript is a good fit for Nature Communications. Several of the points I will discuss below are blatantly incorrect or misleading. Without rectifying these mistakes, I cannot endorse publication.

At no point are you determining the crystallographic axes

The methodology of the manuscript repeatedly uses language that implies the method is capable of determining the crystallographic axes of the involved crystals. This is incorrect and misleading language. At no point are the crystallographic axes identified. The authors use their method to determine what are presumably crystallographic facets of an unknown termination. This is the whole backing premise of why they use the "Rotation 30 technique." This discussion needs to be clarified and correct language must be used before I can support publication.

hBN flipping

This point was brought up in the previous round of discussion but I feel is still not sufficiently addressed. Again, the whole premise of trying to make devices using the same surface of hBN is predicated on the hBN being AA' stacking. If hBN were AA, then the choice of surface does not matter. The authors do not address the fact that flipping of hBN is done to preserve surface termination and requires the exact same number of atomic layers between the sections of hBN that are used in the stack. This is certainly not the case in all of the presented devices. For example, in Figure 3 panels d and f, the two sections of hBN are different colors and therefore cannot be assumed to have the same surface termination. One cannot assume that two disjoint sections of hBN have the same number of layers. The only way of asserting that there are exactly the same number of hBN crystals is to cut a hBN flake that has been verified to be atomically flat (using a technique like AFM) in half. One can only optically resolve thickness hBN flakes on the level of several nanometers. Without verifying the crystal is atomically flat, there is no reason to use this flipping technique and it should not be advertised as a way to control the proximate surfaces of hBN to the graphene layer.

While I appreciate the sentiment of the "Golden rules," the third rule is not a meaningful statement and is certainly not broadly applicable. The whole idea of this third rule is essentially self-evident from the start. In order to stack crystals with angular certainty of a certain amount, it is evident that one must first know the orientation of the reference edges to that degree of certainty (assuming the stacking process does not insert additional uncertainty of a comparable or larger order which is not always the case, something that is not discussed and probably should be). Asserting that one must measure the facet three times is not a meaningful or useful statement. I would suggest either omitting or rephrasing this rule.

Reviewer #2 (Remarks to the Author):

The authors have revised the manuscript according to all the reviewers' comments. The results and discussion have been convincing after the revisions. This manuscript introduces great tips to fabricate doubly aligned h-BN/graphene/h-BN heterostructures, and it is quite useful for researchers working in the research field of 2D materials.

Reviewer #3 (Remarks to the Author):

We appreciate the authors' response to all reviewer comments. The manuscript now appears suitable for publication in Nature Communications.

Point-by-point responses to the reviewers' comments

(Our point-to-point responses are in **red**, and the corresponding changes in the manuscript are in **blue**.)

Report of Reviewer #1 (Remarks to the Author):

The authors have made significant revisions to their manuscript. Overall, it is a more solid manuscript and I still believe the manuscript is a good fit for Nature Communications. Several of the points I will discuss below are blatantly incorrect or misleading. Without rectifying these mistakes, I cannot endorse publication.

Response: We greatly appreciate the reviewer's strong support and helpful suggestions for our work, which have motivated us to further improve our manuscript. In this report, the reviewer points out three instances where our statements seem misleading or incorrect. Those are: (1) Determining the crystallographic axes; (2) Surface determination of hBN flakes during the flip-over step; and (3) the third golden rule. We completely agree with the reviewer and believe these are all simply due to misusing words/sentences in our original text. We have revised all these language issues in the main manuscript, as also described in our point-by-point responses below.

1. At no point are you determining the crystallographic axes. The methodology of the manuscript repeatedly uses language that implies the method is capable of determining the crystallographic axes of the involved crystals. This is incorrect and misleading language. At no point are the crystallographic axes identified. The authors use their method to determine what are presumably crystallographic facets of an unknown termination. This is the whole backing premise of why they use the "Rotation 30 technique." This discussion needs to be clarified and correct language must be used before I can support publication.

Response: We thank the reviewer for the helpful comments. We apologise for the misleading language used here. Yes, we agree with the reviewer that we cannot determine/identify the exact crystallographic axes of the edges, whether zigzag or armchair. What we meant was that we could use the edge to "represent" one of the principal crystallographic axes (PCA) of an unknown termination when using the "Rotation 30 technique". We have clarified the related discussion and corrected the language in the revised manuscript.

2. hBN flipping. This point was brought up in the previous round of discussion but I feel is still not sufficiently addressed. Again, the whole premise of trying to make devices using the same surface of hBN is predicated on the hBN being AA' stacking. If hBN were AA, then the choice of surface does not matter. The authors do not address the fact that flipping of hBN is done to preserve surface termination and requires the exact same number of atomic layers between the sections of hBN that are used in the stack. This is certainly not the case in all of the presented devices. For example, in Figure 3 panels d and f, the two sections of hBN are different colors and therefore cannot be assumed to have the same surface termination. One cannot assume that two disjoint sections of hBN have the same number of layers. The only way of asserting that there are exactly the same number of hBN crystals is to cut a hBN flake that has been verified to be atomically flat (using a technique like AFM) in half. One can only optically resolve thickness hBN flakes on the level of several nanometers. Without verifying the crystal is atomically flat, there is no reason to use this flipping technique and it should not be advertised as a way to control the proximate surfaces of hBN to the graphene layer.

Response: We thank the reviewer for the helpful comments. We agree with the reviewer that the two disjoint sections of the hBN can have a different number of layers, and their top surfaces might not be atomically flat and clean. On the other hand, however, it is highly likely the bottom surfaces of the hBN flakes have the same termination and are much cleaner, as the two hBN flakes are cleaved from the same crystallographic facets of hBN crystal and are not in direct contact with the tape used for the cleaving. Further, the difference in thickness should not affect the termination of the bottom surface. Therefore, the main aim during the flip-over step is to attach the bottom surface of one of the flakes onto the bottom surface of the other flake (both bottom surfaces should have the same termination) instead of combining both top surfaces, as shown in Fig. 3a. We have added the related discussion in the revised manuscript.

3. While I appreciate the sentiment of the “Golden rules,” the third rule is not a meaningful statement and is certainly not broadly applicable. The whole idea of this third rule is essentially self-evident from the start. In order to stack crystals with angular certainty of a certain amount, it is evident that one must first know the orientation of the reference edges to that degree of certainty (assuming the stacking process does not insert additional uncertainty of a comparable or larger order which is not always the case, something that is not discussed and probably should be). Asserting that one must measure the facet three times is not a meaningful or useful statement. I would suggest either omitting or rephrasing this rule.

Response: We thank the reviewer for the helpful comments. We agree with the reviewer that the third Golden Rule should be rephrased. For clarity, we have rephrased the statement expressing this third rule in the manuscript. The main aim of the third rule is to minimize human error as much as possible through multiple measurements of PCA, which is a very important rule that we learnt and derived from our long and extensive experience in conducting this process. In order to enhance alignment precision and success rate, we strongly suggest users to follow this third rule when representing PCA on their optical microscopy systems.

Report of Reviewer #2 (Remarks to the Author):

The authors have revised the manuscript according to all the reviewers' comments. The results and discussion have been convincing after the revisions. This manuscript introduces great tips to fabricate doubly aligned h-BN/graphene/h-BN heterostructures, and it is quite useful for researchers working in the research field of 2D materials.

Response: We thank the referee for the positive appraisal of our revision and strong endorsement for publication.

Report of Reviewer #3 (Remarks to the Author):

We appreciate the authors' response to all reviewer comments. The manuscript now appears suitable for publication in Nature Communications.

Response: We thank the referee for the positive appraisal of our revision and recommendation for publication.

REVIEWERS' COMMENTS

Reviewer #1 (Remarks to the Author):

The authors have made significant revisions to their manuscript and improved it significantly. There are still a few sticking points in the language of the manuscript but at this point I am comfortable accepting the manuscript for publication in Nature Communications.

For example, the section on "Using neighboring graphite edges" could use improvement in the clarity of the language. I believe this section could be improved by explicitly stating the fact that you are using edges which are nominally sharing the same termination as reference points instead of only referencing principal crystallographic axes. Additionally, the sentence at line 226 "we find the PCA of neighboring graphite edges..." still uses incorrect language, but this is a smaller error now. Similarly line 400 also implies that you are directly identifying the crystallographic axes.

The section on the "Flip-over technique" was dramatically helped by the previous edits. I am not sure I agree with the argument that the bottom surfaces of disjoint flakes must share the same termination. This would require the progenitor hBN surface that both flakes were cleaved from to be atomically flat. However, I do strongly agree with the argument for using bottom surfaces as they are free from polymer residue.

Point-by-point responses to the reviewers' comments

(Our point-to-point responses are in **red**, and the corresponding changes in the manuscript are in **blue**.)

Report of Reviewer #1 (Remarks to the Author):

The authors have made significant revisions to their manuscript and improved it significantly. There are still a few sticking points in the language of the manuscript but at this point I am comfortable accepting the manuscript for publication in Nature Communications.

Response: We thank the referee for their positive appraisal of our revisions and for their strong endorsement for publication. In this report, the reviewer points out a few places in the language use for determining the crystallographic axes. We have corrected the language description for this point. These changes are explained in our point-by-point responses below and explicitly described in the revised manuscript.

1. For example, the section on “Using neighboring graphite edges” could use improvement in the clarity of the language. I believe this section could be improved by explicitly stating the fact that you are using edges which are nominally sharing the same termination as reference points instead of only referencing principal crystallographic axes. Additionally, the sentence at line 226 “we find the PCA of neighboring graphite edges...” still uses incorrect language, but this is a smaller error now. Similarly line 400 also implies that you are directly identifying the crystallographic axes.

Response: We thank the Reviewer for the helpful comments. We have changed the language used at line 226 and line 400, by stating that we are using edges which are nominally sharing the same termination as reference points.

2. The section on the “Flip-over technique” was dramatically helped by the previous edits. I am not sure I agree with the argument that the bottom surfaces of disjoint flakes must share the same termination. This would require the progenitor hBN surface that both flakes were cleaved from to be atomically flat. However, I do strongly agree with the argument for using bottom surfaces as they are free from polymer residue.

Response: We thank the Reviewer for the critical comments and agree with us that the bottom surface is better for flip-over technique.